# UNVEILING LINEAR MODE CONNECTIVITY OF RE-BASIN FROM NEURON DISTRIBUTION PERSPECTIVE

## ABSTRACT

In deep learning, stochastic gradient descent (SGD) finds many minima that are functionally similar but divergent in parameter space, and connecting the two SGD solutions will depict a loss landscape called linear mode connectivity (LMC), where barriers usually exist. Improving LMC plays an important role in model ensemble, model fusion, and federated learning. Previous works of re-basin map different solutions into the same basin to reduce the barriers in LMC, using permutation symmetry. It is found that the re-basin methods work poorly in early training and emerge to improve LMC after several epochs. Also, the performances of re-basins are usually suboptimal that they can find permutations to reduce the barrier but cannot eliminate it (or the reduction is marginal). However, there is no unified theory on when and why re-basins will improve LMC above chance, and unveiling the behind mechanism is fundamental to improving re-basin approaches and further understanding the loss landscape and training dynamics of deep learning. Therefore, in this paper, we propose a theory from the neuron distribution perspective to demystify the mechanism behind the LMC of re-basin. In our theory, we use Shannon entropy to depict the uniformity of neuron distributions and derive that non-uniformity (entropy decrease) will result in better LMC after re-basin. In accordance with our theory, we present the following observations, all of which can be aptly explained by our theory. i) The LMC of re-basin changes in various non-uniform initializations. ii) The re-basin's LMC improvement emerges after training due to the neuron distribution change. iii) The LMC of re-basin changes when pruning with different pruning ratios. Building upon these findings, we further showcase how to apply our theory to refine the performances of other neuron alignment methods beyond re-basin, e.g., OTFusion and FedMA.

## 1 INTRODUCTION

Optimization in deep learning is a non-convex problem in high-dimensional space due to its numerous number of parameters and non-linearity of activations. The effectiveness of stochastic gradient descent (SGD) algorithms in deep learning is still an open problem since SGD always robustly finds minima with generalization, but what are all the minima and how are they connected are mysterious (Ainsworth et al., 2022; Vlaar & Frankle, 2022; Lucas et al., 2021).

Visualizing and understanding the loss landscapes of deep neural networks is helpful for demystifying the mechanisms of the SGD process and solutions in deep learning (Li et al., 2018; Nguyen, 2019; Fort & Jastrzebski, 2019). Linear mode connectivity (LMC) depicts the loss landscape of linearly connecting two independent SGD solutions, and it is intriguing to see there usually exist barriers in LMC (Entezari et al., 2021), indicating the two SGD solutions fall into two different loss basins (Ainsworth et al., 2022). LMC plays a significant role in federated learning (Wang et al., 2020b; Li et al., 2020) and fusion-based model ensemble method (Singh & Jaggi, 2020), because they require linear fusion of multiple models, and if the models are not connected in the landscape, the fused model will settle into the bad loss area with poor generalization.

Recent advances conjecture that by taking permutation symmetry (also known as permutation invariance) into account, two SGD solutions can be mapped into the same loss basin (Entezari et al., 2021; Ainsworth et al., 2022), therefore improving LMC and model fusion (Peña et al., 2023; Zhou et al., 2023), and these methods are called "re-basin" (Ainsworth et al., 2022). The re-basin empirically sheds light on the potential of post-matching in improving LMC. However, there is no unified theoretical explanation about how the LMC improvement in re-basin emerges and in which

cases, re-basin can be better above chance. Specifically, the mechanism of re-basin in LMC poses the following pressing questions:

1. Why cannot re-basin improve the LMC at initialization and early training (Ainsworth et al., 2022)?
2. What affects re-basin's LMC after training?
3. In this paper, we observe that pruning can improve re-basin's LMC, but what is the mechanism?

Intuitively speaking, training renders the model increasingly deterministic, making neuron distributions gather around minima and become non-uniform, while pruning restricts parameter distribution to only certain positions, causing non-uniformity as well. The non-uniform neuron distribution may make the re-basin easier. Therefore, we present the following conjecture to explain and understand re-basin's LMC:

**Our Conjecture (informal):** Increasing non-uniformity in neuron parameter distribution leads to the enhancement in linear mode connectivity after applying re-basin.

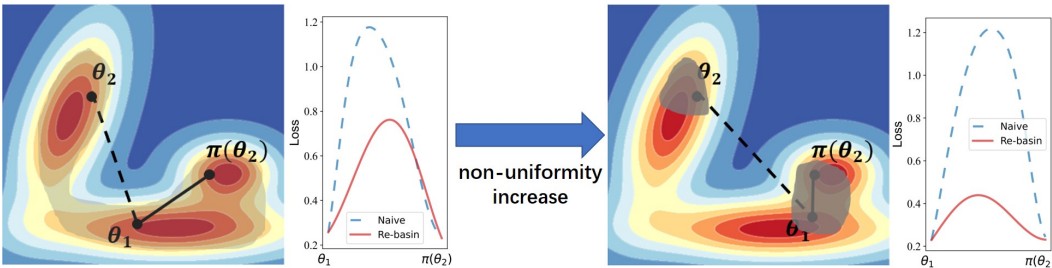

Figure 1: **How increasing the non-uniformity of neuron parameters can enhance re-basin's linear mode connectivity.** The shadow areas represent possible distribution of parameters after training. The higher the non-uniformity of neuron parameters, the narrower the region where the parameters are most likely distributed. Such non-uniformity facilitates easier matching between parameters, and consequently, after the re-basin process, the linear mode connectivity between models is enhanced.

The intuition of our conjecture is shown in Figure 1. In this paper, we provide a formal definition of neuron distribution non-uniformity and discuss its relationship with re-basin's LMC, both theoretically and empirically. We validate our conjecture and explore its practical implications for model fusion. In summary, the main contributions of this paper are as follows:

1. We first introduce a theoretical framework based on neuron distribution non-uniformity to analyze the LMC of re-basin. We use Shannon entropy to depict the uniformity of neuron distributions and derive that non-uniformity (entropy decrease) will result in better LMC after re-basin.

2. Empirically, our theory is justified under three distinct scenarios: i) under different non-uniform initializations; ii) before and after training; and iii) pruning. We discover and highlight that pruning can enhance the effects of re-basin.

3. By applying our theory and findings to other neuron alignment techniques, such as OTFusion (Singh & Jaggi, 2020) and FedMA (Wang et al., 2020b), we improve the accuracy of the fused models, which showcases the prospects of our theory.

## 2 BACKGROUND

In this section, we provide the basic backgrounds and definitions regarding linear mode connectivity, neuron distribution entropy, and the re-basin algorithm in Ainsworth et al. (2022) for finding permutation symmetry to improve LMC.

### 2.1 DEFINITIONS

Let $N$ be the number of training samples, $\mathbb{X} = [\boldsymbol{x}_1, ..., \boldsymbol{x}_N]^T \in \mathbb{R}^{N \times d_{\text{in}}}$ and $[\boldsymbol{y}_1, ..., \boldsymbol{y}_n]^T \in \mathbb{R}^{N \times d_{\text{out}}}$ be inputs and labels of training dataset respectively with $\boldsymbol{x}_i \in \mathbb{R}^{d_{\text{in}}}$ and $\boldsymbol{y}_i \in \mathbb{R}^{d_{\text{out}}}$. Let $f_{\boldsymbol{\theta}}(\boldsymbol{x})$

be a function represented by a neural network with parameter vector $\boldsymbol{\theta} \in \mathbb{R}^M$ which includes all parameters of the the neural network and $L(f_{\boldsymbol{\theta}}(\mathbb{X}), \mathbb{Y})$ be the convex loss function measures the difference between the neural network's predictions and the labels of dataset $\{\mathbb{X}, \mathbb{Y}\}$. Let $L(\boldsymbol{\theta}) = L(f_{\boldsymbol{\theta}}(\mathbb{X}), \mathbb{Y})$ as a function of parameter vector $\boldsymbol{\theta}$ on space $\mathbb{R}^M$ for fixed dataset. We aim to explore the re-basin process, which finds permutation symmetry to improve the linear connectivity of two parameters $\boldsymbol{\theta}_1$ and $\boldsymbol{\theta}_2$. The linear connectivity is depicted by the barrier in the loss landscape between $\boldsymbol{\theta}_1$ and $\boldsymbol{\theta}_2$ along a linear path, defined as follows.

**Definition 2.1** *(Loss barrier (Entezari et al., 2021)) Given two parameters $\boldsymbol{\theta}_1, \boldsymbol{\theta}_2$, the loss barrier $B(\boldsymbol{\theta}_1, \boldsymbol{\theta}_2)$ is defined as the highest difference between the loss occurred when linearly connecting two parameters $\boldsymbol{\theta}_1, \boldsymbol{\theta}_2$ and linear interpolation of the loss values at both of them:*

$$B(\boldsymbol{\theta}_1, \boldsymbol{\theta}_2) = |\sup_{\alpha}[L(\alpha\boldsymbol{\theta}_1 + (1-\alpha)\boldsymbol{\theta}_2)] - \alpha L(\boldsymbol{\theta}_1) - (1-\alpha)L(\boldsymbol{\theta}_2)|. \tag{1}$$

In this work, we mainly use the neural distribution entropy in each layer to study the LMC of re-basin. Here, we provide the formal definition of non-uniformity using the neural distribution entropy.

**Definition 2.2** *Neuron Distribution Entropy Consider an arbitrary network $f_{\boldsymbol{\theta}}(\boldsymbol{x})$ and an arbitrary layer $i$ of network $f_{\boldsymbol{\theta}}(\boldsymbol{x})$. If neurons in layer $i$ all follows the probability distribution $w \sim P$, then the neuron distribution entropy of layer $i$ is defined as the discrete estimation Shannon entropy $H_{\triangle}(P)$ of the probability distribution.*

**Remark 2.3** *$H_{\triangle}(P)$ means discretizing each 1-dimensional continuous random variable component into $2^N$ intervals and calculating discrete entropy. For smooth distributions, its relationship with continuous entropy $H(P) = -\int_{\mathbb{R}^d} p(\boldsymbol{x}) \log p(\boldsymbol{x}) d\boldsymbol{x}$ is $H_{\triangle}(P) = d \cdot \triangle + H(P)$ for $d$-dimension random vector, while for discrete distributions with defined values, its value is $H_{\triangle}(P)$. Here $d$ is the dimension of $w$ and $\triangle$ is the length of the discrete interval. The reason for employing this discrete estimation of Shannon entropy can be found in Appendix B. In this paper, we also define $\tilde{H} = \frac{H}{d}$ as the average entropy for each element of a neuron. And for continuous distribution, we sometimes use continuous entropy $H$ for simplicity, because the difference between $H$ and $H_{\triangle}$ is a constant $\triangle$*

In the subsequent sections, for simplicity and without losing generality, we primarily use multi-layer fully connected neural networks for theoretical study. Here we provide a formal definition of an $L$-layer fully connected neural network. Let $n_i$ be the number of neurons at layer $i$ and $\boldsymbol{W}_i \in \mathbb{R}^{n_i \times n_{i-1}}$ and $\boldsymbol{b}_i \in R^{n_i}$ be the weight matrix and biases at layer $i$ respectively, where $n_L = d_{out}$ and $n_0 = d_{in}$. Let $\sigma : \mathbb{R} \to \mathbb{R}$ be a continuous activation function following Lipschitz assumption, which is applied to vector or matrix element-wise. Each layer is represented by a function $f^i_{(\boldsymbol{W}_i, \boldsymbol{b}_i)}$ and the output $\boldsymbol{z}_i$ of the $i$-th layer is called the activations of the $i$-th layer, where $\boldsymbol{z}_{i+1} = f^i_{(\boldsymbol{W}_i, \boldsymbol{b}_i)}(\boldsymbol{z}_i)$. The neural network can be represented recursively as

$$\begin{cases} \boldsymbol{y} = f^L_{(\boldsymbol{W}_L, \boldsymbol{b}_L)}(\boldsymbol{z}_{L-1}) = \boldsymbol{W}_L \boldsymbol{z}_{L-1}, \\ \boldsymbol{z}_{i+1} = f^L_{(\boldsymbol{W}_{i+1}, \boldsymbol{b}_{i+1})}(\boldsymbol{z}_i) = \sigma(\boldsymbol{W}_{i+1}\boldsymbol{z}_i + \boldsymbol{b}_{i+1}), 0 < i < L, \\ \dots, \\ \boldsymbol{z}_1 = f^L_{(\boldsymbol{W}_1, \boldsymbol{b}_1)}(\boldsymbol{x}) = \sigma(\boldsymbol{W}_1 \boldsymbol{x} + \boldsymbol{b}_1). \end{cases} \tag{2}$$

For the analysis in the following sections, we ignore the biases $\boldsymbol{b}_i$ for simplicity, as it can be added to the weight matrix $\boldsymbol{W}_i$ through small adjustment.

## 2.2 Preliminary of Git Re-basin

For two arbitrary solutions $\boldsymbol{\theta}_1, \boldsymbol{\theta}_2$ found by SGD, there is almost no linear connectivity between them (Draxler et al., 2018; Garipov et al., 2018), while Ainsworth et al. (2022); Entezari et al. (2021) demonstrate by applying a layer-wise permutation $\pi$ to the parameter $\boldsymbol{\theta}_1$ and leveraging the permutation invariance of neural network, it induces linear connectivity between the parameters $\boldsymbol{\theta}_2$ and the equivalent parameter $\pi(\boldsymbol{\theta}_1)$ in practice. Here "equivalent" means that for any input $\boldsymbol{x}$, the output $f_{\pi(\boldsymbol{\theta}_1)}(\boldsymbol{x})$ and $f_{\boldsymbol{\theta}_1}(\boldsymbol{x})$ are equal. It is indicated that the permutation invariance of fully connected networks and that of the other kinds (e.g., convolutions) are similar (Ainsworth et al., 2022). Permutation invariance is shown by applying arbitrary permutation matrix $\boldsymbol{P}_i \in \mathbb{R}^{n_i \times n_i}$ to the weight matrix $\boldsymbol{W}_i, \boldsymbol{W}_{i+1}$ of $i$-th layer and $i+1$-th layer, the new neural network with new weight matrices $\boldsymbol{W}'_i = \boldsymbol{P}_i \boldsymbol{W}_i, \boldsymbol{W}'_{i+1} = \boldsymbol{W}_{i+1}\boldsymbol{P}_i^T$ is equivalent to the original neural network such that

$$\boldsymbol{W}'_{i+1}\boldsymbol{z}'_i = \boldsymbol{W}_{i+1}\boldsymbol{P}_i^T \boldsymbol{z}'_i = \boldsymbol{W}_{i+1}\boldsymbol{P}_i^T \sigma(\boldsymbol{P}_i \boldsymbol{W}_i \boldsymbol{z}_{i-1}) = \boldsymbol{W}_{i+1}\sigma(\boldsymbol{W}_i \boldsymbol{z}_{i-1}) = \boldsymbol{W}_{i+1}\boldsymbol{z}_i. \tag{3}$$

After applying a layer-wise permutation $\pi = (\boldsymbol{P}_1, ..., \boldsymbol{P}_{L-1})$ and repeating the above process, the permuted neural network have parameter $\pi(\boldsymbol{\theta}) = (\boldsymbol{P}_1 \boldsymbol{W}_1, \boldsymbol{P}_2 \boldsymbol{W}_2 \boldsymbol{P}_1^T, ..., \boldsymbol{W}_L \boldsymbol{P}_{L-1}^T)$.

Finding the optimal permutation $\pi$ that minimizes the loss barrier $B(\pi(\boldsymbol{\theta}_1), \boldsymbol{\theta}_2)$ is NP-hard, while the prior work Ainsworth et al. (2022) summarizes three effective methods for searching for the optimal permutations and these methods are called "re-basin". We primarily employ the weight matching method as the studied re-basin method, as it aligns closely with our theoretical analysis and it has broad applications. The weight matching method is shown in subsection A.2 and the objective of the weight matching method is to find permutation solving problem $\min_\pi ||\pi(\boldsymbol{\theta}_1) - \boldsymbol{\theta}_2||_2$, which promotes LMC under the assumption that "two neurons are associated when they have close values" (Ainsworth et al., 2022). In our analysis, we will also show this assumption is meaningful.

## 3 THEORETICAL RESULTS

In this subsection, we introduce a theoretical framework for analyzing multi-layer fully connected networks and relate the LMC of re-basin to the random Euclidean matching problems (Theorem 3.1). Using bounds (Theorem 3.2) from the random Euclidean matching problems (Goldman & Trevisan, 2022; Ambrosio & Glaudo, 2019; Goldman & Trevisan, 2021), we demonstrate the correlation between LMC and the entropy of neuron distribution (Theorem 3.3), which serves as the foundation for our subsequent analysis of the role of the neuron distribution entropy in the LMC of re-basin.

**Theorem 3.1 (Relation between Random Matching Problem and Linear Mode Connectivity )** *If each row $\boldsymbol{w}_{j,:}^{(i)}$ of the weight matrix $\boldsymbol{W}_i$ of layer $i$ follows distribution $\mathbb{R}^{n_{i-1}} \ni \mathbf{w} = (\mathrm{w}_1, ..., \mathrm{w}_{n_{i-1}}) \sim P$ i.i.d. with $b_i \triangleq \sqrt{\sum_{j=1}^{n_{i-1}} Var(\mathrm{w}_j)}$, and the input of the neural network $\boldsymbol{x}$ is bounded $||\boldsymbol{x}||_2 < b_x$, then for any $\delta > 0$, with probability $1 - \delta$,*

$$\sup_{\alpha \in [0,1]} |f_{\alpha\boldsymbol{\theta}_1 + (1-\alpha)\boldsymbol{\theta}_2}(\boldsymbol{x}) - \alpha f_{\boldsymbol{\theta}_1}(\boldsymbol{x}) - (1 - \alpha) f_{\theta_2}(\boldsymbol{x})| \leq B_{L-1} b_x. \tag{4}$$

*And $B_{L-1}$ is bounded by the following recursive equations*

$$\begin{aligned} B_{i+1} &= \tilde{O}(n_{i+1}^{\frac{1}{2}} b_{i+1}(B_i + D_{i-1})); \\ D_i &= \tilde{O}(n_{i+1}^{\frac{1}{2}} b_{i+1} D_{i-1} + D_E(\boldsymbol{W}_{i+1}^{(1)}, \boldsymbol{W}_{i+1}^{(2)}) \Pi_{j=1}^i n_j^{\frac{1}{2}} b_j b_x); \\ B_1 &= 0, D_0 = \tilde{O}(D_E(\boldsymbol{W}_1^{(1)}, \boldsymbol{W}_1^{(2)}) b_x); \end{aligned} \tag{5}$$

*where $D_E(\boldsymbol{W}_i^{(1)}, \boldsymbol{W}_i^{(2)}) = E_{\boldsymbol{W}_i^{(1)}, \boldsymbol{W}_i^{(2)}} \min_{\boldsymbol{P} \in \mathbb{S}_{n_i}} ||\boldsymbol{P}\boldsymbol{W}_1 - \boldsymbol{W}_2||_2$ as the expectation of minimum distance after permutation and $\mathbb{S}_{n_i}$ is the set of permutation matrices with size $n_i \times n_i$. The proof can be found in subsection D.1*

Although the form of this theorem is relatively complex, it indeed has a theoretical value which we will show in Appendix C with examples. In Appendix C, some propositions related to LMC are deduced by Theorem 3.1, including an extended result of the Theorem 3.1 in Entezari et al. (2021).

**Lemma 3.2 (Relation between Neuron Distribution Entropy and A Bound of Random Matching)** *Consider matrices $\boldsymbol{W}_1, \boldsymbol{W}_2 \in \mathbb{R}^{n_1 \times n_2}$ whose rows are i.i.d. random vector of arbitrary Hölder continuous probability density $\rho$ on a bounded open set $\Omega$, for large $n_2 > 2$, the bound for $D(\boldsymbol{W}_1, \boldsymbol{W}_2)$ becomes $D(\boldsymbol{W}_1, \boldsymbol{W}_2) \leq c_\delta n_1^{\frac{1}{2} - \frac{2}{n_2}} e^{c\tilde{H}(\rho)}$ with probability $1 - \delta$ and constant $c$. The proof can be found in subsection D.2*

Theorem 3.1 elucidates the relationship between the random matching problem and LMC, while Theorem 3.2 demonstrates the relevance of neuron distribution entropy to the bound of the random matching problem. By combining Theorem 3.2 and Theorem 3.1, we derive the principal theoretical finding of this paper, which reflects the correlation between neuron distribution entropy and LMC.

**Theorem 3.3 (Effect of Neuron Distribution Entropy in Linear Mode Connectivity)** *The other conditions are identical to those of Theorem 3.1 and then the bound $B_{L-1} b_x$ is $\tilde{O}(f(n_1, ..., n_{L-1}) \max_i e^{c\tilde{H}(\rho_i)})$, where $\rho_i$ is the probability density of layer $i$, and $f(n_1, ..., n_{L-1})$ is a polynomial function of $n_1, ..., n_{L-1}$.*

This theorem presents the primary finding of this paper, illustrating that the increasing non-uniformity (which means a reduction of neuron distribution entropy $\tilde{H}_\Delta(\rho)$) within neural networks can reduce the loss barrier and augment the LMC of re-basin at an exponential rate. The conditions required to satisfy naturally hold when calculating the LMC of re-basin at initialization time, using common initialization methods (Glorot & Bengio, 2010; He et al., 2015). If the training process follows the constraints $A1 \sim A4$ outlined in Mei et al. (2018; 2019), this theorem still holds both during and after the training process. In addition, our experiments demonstrate that in other cases, the proposition that neuron distribution can enhance the LMC of re-basin also holds true. This conclusion will be further validated through three scenarios in the subsequent section, from which its applications will be derived.

## 4 ANALYSIS IN PRACTICE

In this section, we will explore the impact of neuron distribution entropy on the LMC of re-basin in three scenarios, where the neuron distribution entropy and the LMC of re-basin change together. The following scenarios are validated by our theory: (1) the influence of varying non-uniform initializations with different entropy on the LMC of re-basin at initialization, (2) the increase in neuron distribution entropy before and after the training process enhances the LMC of re-basin, and (3) pruning algorithms enhance the LMC of re-basin. Among them, enhancing non-uniformity before and after the training process provides an explanation for the phenomenon elucidated in Ainsworth et al. (2022) that the LMC of re-basin enhances after training, and our finding that pruning algorithms enhancing the LMC of re-basin of neural networks provides a possible practical application for our theoretical framework, which leads to the application of our theory in model fusion and federated learning in section 5.

### 4.1 NEURON DISTRIBUTION ENTROPY AT INITIALIZATION

This scenario is used to validate our theory under the simplest conditions. We consider different initialization distributions for parameters, and then randomly select two parameter points based on that distribution. We compute the value of their loss barrier after the re-basin process as an indicator of linear mode connectivity. Then we compare the trends of the loss barrier value and the entropy of the initialization distribution and validate the relationship between them. The following initialization scheme is primarily used to lead to different neuron distribution entropy: components of the neurons follow a normal distribution i.i.d, and by altering the standard

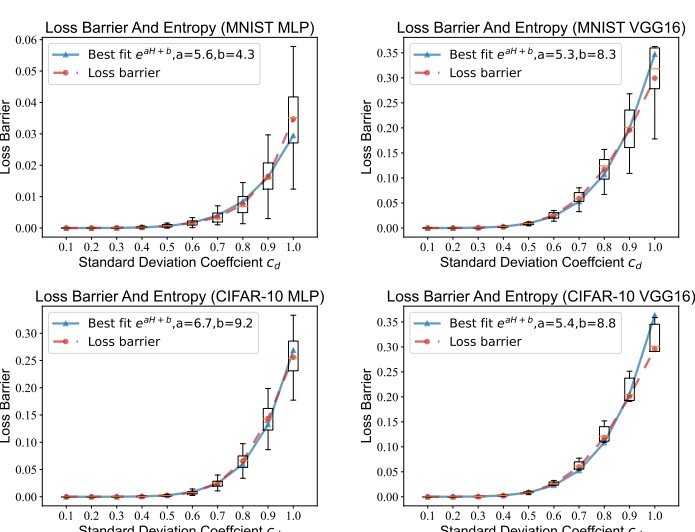

Figure 2: **Neuron distribution entropy and LMC of re-basin at different initialization**. Here we use function $L = e^{aH+b}$ to fit the values of entropy $H$ and loss barrier $L$. The blue curve in the graph represents the best-fit line. (30 models are used for statistics)

deviation, we achieve varying degrees of non-uniformity. Here the standard deviation is set to $c_d\sigma_{\text{He}}$, where $\sigma_{\text{He}}$ is the standard deviation for the initialization in He et al. (2015); and $c_d$, called standard deviation coefficient, its changes resulting to different entropy. To obviate the influence of the absolute magnitude of the loss itself, for scheme (1), we normalize the loss barrier by the model's intrinsic loss, namely $\frac{B(\boldsymbol{\theta}_1, \boldsymbol{\theta}_2)}{(L(\boldsymbol{\theta}_1)+L(\boldsymbol{\theta}_2))/2}$. After obtaining the data of loss barrier $L$ and neuron distribution entropy $H$ at initialization, we fit the function $L = e^{aH+b}$ with the data and find that $L$ increases exponentially with the rise of $H$, which is consistent with Theorem 3.3. The results are shown in Figure 2.

### 4.2 THE CHANGES OF NEURON DISTRIBUTION ENTROPY AFTER TRAINING

Now, we show the decrease in neuron distribution entropy before and after the training process enhances linear mode connectivity. Take a wide two-layer fully connected network for example. Initially, the parameter vectors of neurons in the hidden layer are assumed to follow some distribution $\rho_0$ i.i.d, and after $k$ training steps and assuming conditions A1-A4 in Mei et al. (2018; 2019) hold, these parameter vector of neurons still follow a distribution $\rho_k$ i.i.d., which is given by $\rho_k = \rho(\sum_{i=0}^{k} s_i)$ and $\partial_t \rho = 2\xi(t)\nabla_\theta \cdot (\rho_t \nabla_\theta \Psi_\lambda(\theta; \rho))$, where $s_k$ is the step size at the $k$th step and see (Mei et al., 2018; 2019) for definitions of other symbols. Training a neural network is an iterative refinement process, causing the entropy of $\rho$ to reduce over time when the parameter distribution gradually converges to the lowest points on the loss surface. According to Theorem 3.3, a decrease in the entropy $\tilde{H}(\rho)$ of $\rho$ will result in a decrease in the bound of the loss barrier. Consequently, the linear mode connectivity strengthens as training continues.

To validate the process above, we conduct experiments using the First and Third Degree Polynomial dataset and the single-input, single-output MLP with two hidden layers in Von Oswald et al. (2019); Peña et al. (2023). Our analysis focuses on comparing the distribution of neuron parameters in the first hidden layer, the entropy, and

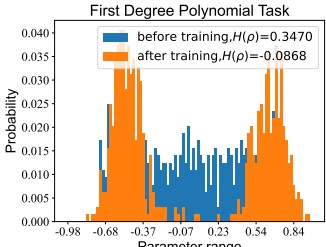 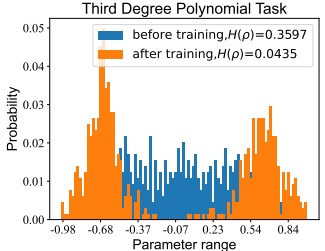

Figure 3: **Changes in neuron distribution before and after training.** The first layer of a single-input MLP (64 models are used for statistics).

the loss barrier before and after training. As a result, Figure 3 shows the changes in the distribution of neuron coefficients in the first layer of the neural network before and after training, which shows a noticeable reduction in the entropy of the neuron distribution. On the First Degree Polynomial, the training loss barrier and test loss barrier decrease from 0.1523 and 0.1560 to 0.0460 and 0.0538, respectively. For the Third Degree Polynomial, these values decrease from 0.1434 for both to 0.0189 and 0.0264, respectively.

### 4.3 DECREASE OF NEURON DISTRIBUTION ENTROPY WHEN PRUNING

Intuitively speaking, pruning causes certain parameters of neurons to become the value zero, making the neuron's values more deterministic, thereby reducing the entropy of the neuron distribution. Therefore, according to Theorem 3.3, pruning should lead to an enhancement in the network's linear mode connectivity after re-basin. The statement of the decline in entropy of the neuron distribution after pruning can be proved formally by Lemma 4.1. In Lemma 4.1, before pruning, distributions of the individual neuron components are described by random variables $x_1, ..., x_n$ and after pruning with a ratio of $r/n$, the components of neurons can be described by random variables $y_{r+1}, ..., y_n$ and $r$ zero components, leading to a reduction in the entropy.

**Lemma 4.1** *Let $x_1, x_2, ..., x_n$ be $n$ i.i.d. random variables with continuous cumulative distribution function $F(x)$ and probability density function $f(x)$. Let $x_{i:n}$ denotes the $i$-th order statistics of all the $n$ random variables and $y_{r+1}, ..., y_n$ be the order statistics $x_{r+1:n}, ..., x_{n:n}$ without order. Then the entropy $H_\Delta(0, 0, ..., 0, y_{r+1}, ..., y_n)$ is less than $H_\Delta(x_1, ..., x_n)$, where $H_\Delta(0, 0, ..., 0, y_{r+1}, ..., y_n)$ denotes the approximation discrete entropy of the union distribution of random variables $y_{r+1}, ..., y_n$ and $r$ zero-valued deterministic random variables and $H_\Delta(x_1, ..., x_n)$ similarly.*

Then we empirically validate the conclusion that pruning can enhance the linear mode connectivity of models. To maintain consistency with Lemma 4.1, we apply a consistent pruning rate using the local unstructure pruning method across all layers (Li et al., 2016; Han et al., 2015). We employ three pruning strategies: (1) **Only Pruning** (pruning applied only after training). (2) **Lottery Ticket Hypothesis** (Frankle & Carbin, 2018). (3) **Pruning Followed by Fine-tuning**. Details of the testing results of our pruning process are shown in subsection A.3. After pruning, we conduct a re-basin operation on models, calculate and observe their linear mode connectivity, and find that pruning increases linear mode connectivity. In our experiments, we train MLP, VGG, and ResNet neural networks on both MNIST and CIFAR-10 datasets. In our preliminary experiments, we prune the

Table 1: **Barrier loss comparison with re-basin and our pruning method under different tasks.**
Linear mode connectivity of re-basin is enhanced after pruning.

| Stage | Method | VGG16 CIFAR-10 | VGG16 MNIST | ResNet20 CIFAR-10 | ResNet20 CIFAR-100 | MLP CIFAR-10 |
|---|---|---|---|---|---|---|
| Train | Re-basin | 0.4734 | 1.7772 | 0.4326 | 2.1639 | 0.1925 |
| | **Ours** | **0.3657** | **1.2591** | **0.4154** | **1.6108** | **0.1552** |
| Test | Re-basin | 0.3205 | 1.7446 | 0.2464 | 1.6172 | 0.1048 |
| | **Ours** | **0.2841** | **1.2326** | **0.2294** | **1.2239** | **0.0732** |

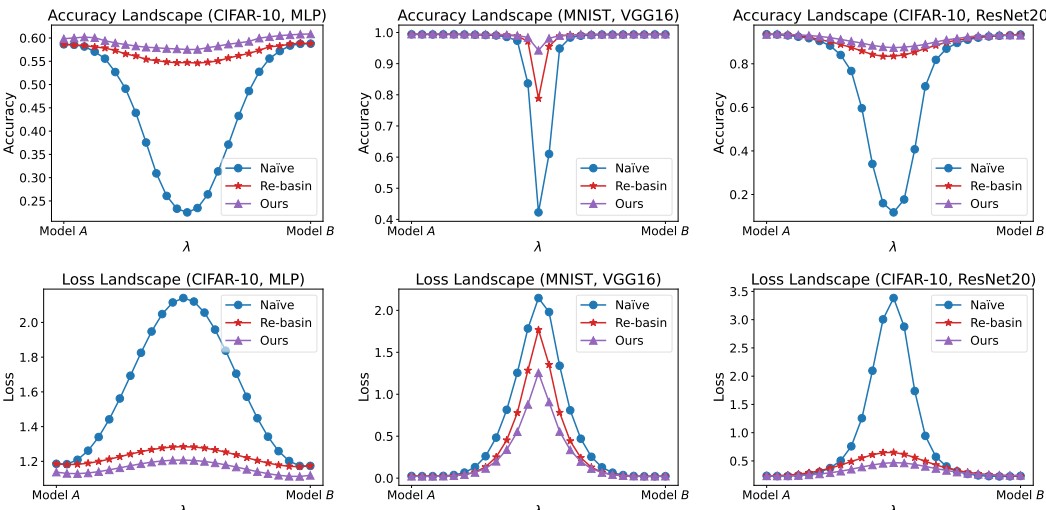

Figure 4: **The loss and accuracy landscape after re-basin for MLP, VGG16, ResNet20 on different Dataset.** Linear mode connectivity is enhanced after our pruning strategy. More results are shown in subsection A.4.

models with varying pruning rates and compare the results between the linear mode connectivity of the pruned models and the linear mode connectivity of their non-pruned counterparts after re-basin. It is found that the pruning strategies **Pruning Followed by Fine-tuning** can lead to the best linear mode connectivity and we take it as our pruning+re-basin method. The results of our pruning+re-basin method can be found in Figure 4 and Table 1. It can be observed that pruning leads to a significant reduction in the loss barrier, indicating an enhancement in linear mode connectivity. The results of **Only Pruning** and **Lottery Ticket Hypothesis** and discussion of the reasons for their failure are shown in subsection A.3.

We also observe that during the pruning ratio increases, LMC first increases and then drops Figure 5. This phenomenon could be explained as follows. At a low pruning rate, even though the model's performance decreases after pruning, it can be restored to its original performance through fine-tuning. At the same time, the neuron distribution entropy is lower, as a result, the loss barrier is reduced, so the performance of the merged model is better. However, as the pruning rate increases, the performance loss of the model cannot be restored through fine-tuning, leading to a continuous decline in model performance.

## 5 EXPERIMENT AND APPLICATION

In this section, we pull out our conclusion that "pruning can improve linear mode connectivity" to other neuron alignment methods than weight matching re-basin method by experiments, including OTFusion (Singh & Jaggi, 2020) and FedMA (Wang et al., 2020b), and show that our conclusion can be applied to abundant applications, including scenarios with multiple models and heterogeneous data.

**Experiments on OTFusion**   To prove our method transferable, we apply our pruning strategies to improve the OTFusion method (Singh & Jaggi, 2020). OTFusion develops an optimal-transport-based

Table 2: **The accuracy of the fused model after OTFusion with pruning.** $m$: the number of samples used in OTFusion. The number of models is 2, lr = 0.004 (SGD), batch size = 128/1000 for VGG and 64/1000 for ResNet. The fine-tuning learning rate is set as 1e-3, and the corresponding epoch is 30. (Singh & Jaggi, 2020).

| Networks | Pruning Rate | 0%(baseline) | 10% | 20% | 30% | 40% | 50% | 60% |
|---|---|---|---|---|---|---|---|---|
| | Simple Pruning | 85.44 | 85.02 | 85.09 | 84.99 | 83.91 | / | / |
| VGG | Pruning with One-shot Fine-tuning | 85.44 | 86.92 | 86.44 | 85.57 | 84.95 | / | / |
| | Pruning with Fine-tuning Multiple Times | 85.44 | 86.67 | 86.21 | 85.9 | 85.09 | / | / |
| | Simple Pruning | 67.19 | 69.02 | 68.96 | 69.02 | 68.97 | 68.76 | 64.39 |
| ResNet50/m=100 | Pruning with One-shot Fine-tuning | 67.19 | 69.83 | 70.66 | 71.44 | 71.61 | 71.44 | 67.05 |
| | Pruning with Fine-tuning Multiple Times | 67.19 | 68.66 | 70.65 | 71.14 | 71.93 | 71.52 | 66.74 |
| | Simple Pruning | 68.8 | 68.31 | 69.16 | 70.13 | 70.75 | / | / |
| ResNet50/m=200 | Pruning with One-shot Fine-tuning | 68.8 | 68.04 | 68.06 | 70.07 | 69.49 | / | / |
| | Pruning with Fine-tuning Multiple Times | 68.8 | 68.46 | 68.58 | 70.08 | 69.37 | / | / |

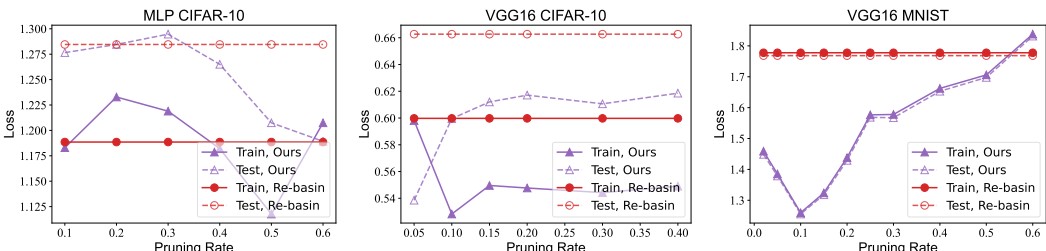

Figure 5: **Changes in loss of the linearly connected model with different pruning rates, comparing re-basin and our pruning+re-basin method.**

strategy to achieve neuron alignments for enhancing the ensembling performance. OTFusion aligns neurons layer by layer and we modify OTFusion by adding pruning operation to each layer before the neuron alignment occur in that layer.

We test our modified OTFusion-pruning method on the dataset CIFAR-10 with network VGG11 and ResNet50. For the pruning process in the modified OTFusion-pruning method, we take 3 different pruning strategies, including (1) Simple Pruning: only implementing local unstructured pruning for each layer; (2) Pruning with One-Shot Fine-Tuning: fine-tuning the model after pruning for certain epochs; and (3) Pruning with Fine-Tuning Multiple Times: repeating pruning and fine-tuning for $n$ times, and each time pruning $\frac{p}{n}$ elements, where $p$ is the whole pruning ratio. In implementation, we prune each layer before the neuron alignment operation in OTFusion.

In Table 2, it is shown that the modified OTFusion approach with pruning will have enhanced accuracy of the fused model, which is consistent with our observation in subsection 4.3 that the connectivity of the re-basin method benefits from pruning.

As an efficient method, our three pruning strategies steadily surpass the baseline with a low pruning rate. In the Table 2, we notice that fine-tuning strategies (one-shot, multi-times) work better than simple pruning in VGG and ResNet with $m = 100$ and they are better than the result of $m = 200$. In this format, the accuracy of pruning methods goes through the rise to a peak, then drops down, which exposes a similar law to our main experiment's Theorem 4.1 study of effect pruning.

**Experiments on FedMA**  Our pruning+neuron alignment strategy works on the very basic units (neurons and layers) and ensembling model, the intuition tells us we can use it on federated learning. FedMA (Wang et al., 2020a) constructs the shared global model in a layer-wise manner by matching neurons and activations to improve the model aggregation in federated learning. We implement our pruning+neuron alignment strategy by incorporating pruning before the FedMA process in federated learning. It is observed that this strategy can promote the global model generalization. Due to the space limit, we include the results in subsubsection A.1.3.

## 6 RELATED WORK

**(Linear) Mode connectivity** Freeman & Bruna (2016); Garipov et al. (2018); Draxler et al. (2018); Nguyen (2019) demonstrate that after training with the SGD algorithm, any two minima on the loss surface can be connected by a curve with low loss. This phenomenon is termed "mode connectivity". However, direct low-loss connections are typically absent. Another study by Entezari et al. (2021) hypothesizes that due to the permutation invariance of neural networks, the minima found by SGD could be directly connected by a low-loss line after an appropriate permutation, which is referred to as "linear mode connectivity", and its experiments provide evidence supporting this hypothesis. Ainsworth et al. (2022) concludes three neuron alignment methods to identify permutations facilitating linear mode connectivity between minima, which is termed 're-basin'. Subsequently, Peña et al. (2023) extends the Gumbel-Sinkhorn method to achieve a differentiable re-basin for broader applications. Mode connectivity offers both algorithmic inspiration and theoretical support for a wide range of applications, including continual learning (Mirzadeh et al., 2020; Lubana et al., 2022), model ensemble (Benton et al., 2021; Liu et al., 2022), pruning (Frankle et al., 2020), and adversarial robustness (Zhao et al., 2020). Among them, Frankle et al. (2020) is similar to the application of our work, but it uses linear mode connectivity as a measure of stability for pruning while our approach integrates pruning techniques into the re-basin process, enhancing the linear mode connectivity between minima.

**Model Fusion** Model fusion is crucial in federated learning since several local models need to be fused into one global model on the server. Bayesian nonparametric framework (Yurochkin et al., 2019) is utilized for better neuron alignment in federated learning. FedMA (Wang et al., 2020b) further extends this Bayesian nonparametric framework by considering permutation invariance and using a layer-wise manner. Additionally, the method OTFusion (Singh & Jaggi, 2020) utilizing optimal transport is devised to improve neuron alignment in model fusion, and it can realize one-shot knowledge transfer. Graph matching is also used in neuron matching for better model fusion (Liu et al., 2022). In federated learning, ensemble distillation methods (Lin et al., 2020; Chen & Chao, 2021) are proposed to improve the global model after model fusion. Besides, it is found that global weight shrinking is beneficial to model fusion by setting the sum of fusion weights smaller than one (Li et al., 2023).

**Random Bipartite Matching Problem** The Random Bipartite Matching problem primarily examines statistic information of $L_{MBM}$, the smallest possible sum of distances between corresponding vectors from two identically distributed random vector sets, after any possible permutation. Steele (1981); Boutet de Monvel & Martin (2002); Goldman & Trevisan (2021) provide a formula for the asymptotic behavior of expectation of $L_{MBM}$ as the number of vector elements increases, along with concentration inequalities for uniformly distributed random vectors in square region While Ledoux (2019); Ledoux & Zhu (2019) discuss this phenomenon for standard Gaussian distribution. Subsequently Ambrosio & Glaudo (2019); Benedetto & Caglioti (2020); Ambrosio et al. (2022) hypothesize and demonstrate the relationship between $L_{MBM}$ and the volume of the probability space for random vectors with more general distributions, indicating a potential relationship between $L_{MBM}$ and characteristics of the distribution. Goldman & Trevisan (2022) details the formula linking the expectation of $L_{MBM}$ for non-uniformly distributed random variables and the Rényi entropy of their distribution. In this paper, we bridge linear mode connectivity with the Random Bipartite Matching problem, leveraging conclusions from the latter to unveil the theoretical relationship between non-uniformity and linear mode connectivity.

## 7 DISCUSSION AND FUTURE WORK

In this paper, we unveil linear mode connectivity after re-basin through neuron distribution and find the role of neuron distribution entropy in linear mode connectivity. We present theoretical findings that establish a relationship between neural distribution entropy and linear model connectivity. Specifically, we assert that a decrease in neural distribution entropy (or an increase in the non-uniformity of neural distribution) can enhance linear mode connectivity. We then empirically validate our theoretical findings across three practical scenarios wherein neural distribution varies, including differing initializations, before and after training, and before and after pruning. During our validation, we find that pruning can improve linear mode connectivity after re-basin. We extend this conclusion to other neuron alignment methods across multiple models and heterogeneous data and improve their performances, demonstrating the practical implications of our theory.

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

# Appendix

In this appendix, we provide the details omitted in the main paper and more analyses and discussions.

- Appendix A: implementation details, including experiment setting, weight matching method of the re-basin method, pruning process, etc.
- Appendix B: some explanations of definitions, experiments,etc.
- Appendix C: some other theoretical results, based on the theoretical analysis in the main text.
- Appendix D: missing proofs for theorems in the main text

## A  IMPLEMENTATION DETAILS

### A.1  EXPERIMENT SETTING

#### A.1.1  EXPERIMENT SETTING FOR SUBSECTION 4.1 AND SUBSECTION 4.2

In subsection 4.1 we change the standard deviation by changing standard deviation coefficient $c_d$ in the set of $\{0.1, 0.2, 0.3, 0.4, 0.5, 0.6, 0.7, 0.8, 0.9\}$ and sample 60 models by the normal distribution with standard deviation $c_d\sigma_{\text{He}}$ for each layer. We test 4 cases, including MLP on MNIST, VGG16 on MNIST, MLP on CIFAR-10 and VGG16 on CIFAR-10. After sample 60 models, we do re-basin operation for pairs of mode $i$ and $i + 1$, $i = 1, ..., 30$ and calculate their train loss barrier $L$. Then we calculate the neuron distribution entropy of each layer using the entropy calculation formula $H = \frac{1}{2}(\ln(2\pi(c_d\sigma_{\text{He}})^2)) + 1$ for normal distribution. Actually, the average entropy $\tilde{H}$ of a neuron is the entropy of its element, for each element is i.i.d. We calculate the maximum value of $\tilde{H}$ of each layer. After getting the data of neuron distribution entropy and loss barrier, we then fit the data by function $L = e^{aH+b}$ and this form is predicted by our theory. In Figure 2, we validate that $L$ increases exponentially with the rise of $H$.

In subsection 4.2, we use the First Degree Polynomial Task and the Third Degree Polynomial Task in Von Oswald et al. (2019); Peña et al. (2023). The reason why we use these tasks is that this task is single-input, and we can easily count the neuron distribution entropy of the first layer, because neurons in the first layer is 1D random vector. Counting entropy for high-dimension random vector is hard and it requires too large amount of sampled models for the complete neuron distribution. The first and third-degree polynomial approximation datasets are $\mathbb{T}_{\text{Pol1}} = \{(x, y)\{y = x + 3, x \in (-4, -2)\}$ and $\mathbb{T}_{\text{Pol3}} = \{(x, y) \mid y = (x - 3)^3, x \in (2, 4)\}$. A small Gaussian noise with distribution $\mathcal{N}(0, 0.05)$ is added to the regression target.

#### A.1.2  EXPERIMENT SETTING FOR SUBSECTION 4.3

Table 3: Experiments settings for different datasets using different models of subsubsection A.1.2

| Model | Dataset | Epoch | Learning Rate | Width Multiplier |
|---|---|---|---|---|
| VGG16 | CIFAR-10 | 100 | 1e-4 | 64 |
| | MNIST | 25 | 1e-2 | 64 |
| ResNet20 | CIFAR-10 | 250 | 1e-1 | 4 |
| | CIFAR-100 | 250 | 1e-1 | 4 |
| MLP | CIFAR-10 | 100 | 1e-1 | / |

In this section that details experiments across various models on different datasets, both the combined pruning and finetuning experiments and the pruning-only experiments maintain consistent experimental settings and hyperparameters. We uniformly employ SGD as the optimizer. The batch size for finetuning is consistently set at 100. For the ResNet20 model (He et al., 2016), the weight decay is set to 1e-4. Hyperparameters for finetuning remain consistent with those of training. For detailed settings and hyperparameters for different tasks, which can refer to Table 3. In the tasks demonstrated in this table, the pruning rate for MLP on CIFAR-10 (Krizhevsky et al., 2009) is set ranging from 0.1 to 0.6 with an increment of 0.1. For ResNet20 on CIFAR-10, the pruning rate ranges from 0.1 to 0.8 with the same increment, whereas on CIFAR-100 it ranges from 0.1 to 0.7. For VGG16 (Simonyan & Zisserman, 2014) on the MNIST dataset (LeCun, 1998), the pruning rates are set as [0.02, 0.05, 0.1, 0.15, 0.2, 0.25, 0.3, 0.4, 0.5, 0.6], on the CIFAR-10 dataset the pruning rates are set as [0.05, 0.1, 0.15, 0.2, 0.4, 0.6].

### A.1.3 EXPERIMENT SETTING FOR SECTION 5

Our experiment have additional supplement materials. Our settings for FedMA can be transformed into the following formula: For the results, we compare to benchmark with prune rate setting 0.1,

Table 4: Experiments settings for FedMA with pruning in section 5, communication round=9, epoch for each train=20

| Model | Dataset | Epoch | Learning Rate | Commute round | Batch Size |
|---|---|---|---|---|---|
| FedMA (with pruning) | CIFAR-10 | 20 | 1e-1(SGD) | 9 | 20 |

Table 5: Experimental results for FedMA with pruning in section 5

| Model | w/o Pruning | Pruning Rate 0.1 | Pruning Rate 0.3 | Pruning Rate 0.5 |
|---|---|---|---|---|
| FedMA (with pruning) | 0.8100 | 0.8142 | 0.8165 | 0.8078 |

0.3, 0.5, and the results can be found in Table 5. The trend of this experiment similar to main pruning experiment and OTFusion-pruning experiment.

### A.2 THE RE-BASIN ALGORITHM, WEIGHT MATCHING

In accordance with the discussion in subsection 2.2, during the re-basing process of weight matching, for the parameters $\boldsymbol{\theta}_1 = (\boldsymbol{W}_1^{(1)}, ..., \boldsymbol{W}_L^{(1)})$ and $\boldsymbol{\theta}_2 = (\boldsymbol{W}_1^{(2)}, ..., \boldsymbol{W}_L^{(2)})$, a layer-wise permutation $\pi = (\boldsymbol{P}_1, ..., \boldsymbol{P}_{L-1})$ is found to apply to $\boldsymbol{\theta}_1$, where $\pi(\boldsymbol{\theta}_1) = (\boldsymbol{P}_1\boldsymbol{W}_1, \boldsymbol{P}_2\boldsymbol{W}_2\boldsymbol{P}_1^T, ..., \boldsymbol{W}_L\boldsymbol{P}_{L-1}^T)$. $\pi$ should achieve the optimal objective of $\min_\pi ||\pi(\boldsymbol{\theta}_1) - \boldsymbol{\theta}_2||_2$. The objective could be transformed into the following formula:

$$\arg\min_{\pi=\{\boldsymbol{P}_i\}} ||\pi(\boldsymbol{\theta}_1) - \boldsymbol{\theta}_2||_2^2 = \arg\max_\pi \pi(\boldsymbol{\theta}_1) \cdot \boldsymbol{\theta}_2$$

$$= \arg\max_{\pi=\{\boldsymbol{P}_i\}} \langle \boldsymbol{W}_1^{(1)}, \boldsymbol{P}_1\boldsymbol{W}_1^{(2)} \rangle_F + \langle \boldsymbol{W}_2^{(1)}, \boldsymbol{P}_2\boldsymbol{W}_1^{(2)}\boldsymbol{P}_1^T \rangle_F + .... \quad (6)$$

$$+ \langle \boldsymbol{W}_L^{(1)}, \boldsymbol{W}_L^{(2)}\boldsymbol{P}_{L-1}^T \rangle_F,$$

where $\langle \boldsymbol{A}, \boldsymbol{B} \rangle_F = \sum_{i,j} A_{i,j}B_{i,j}$ denotes the Frobenius inner product matrices $\boldsymbol{A}$ and $\boldsymbol{B}$. Ainsworth et al. (2022) solves this optimization problem through an iterative approach, called Permutation Coordinate Descent. At each iteration, it focuses on a single permutation matrix $\boldsymbol{P}_l$, which is chosen randomly, while keeping the other permutation matrices unchanged. This helps the optimization problem be simplified into a classical Linear Assignment Problem (LAP):

$$\arg\max_{\boldsymbol{P}_l} \langle \boldsymbol{W}_l^{(1)}, \boldsymbol{P}_l\boldsymbol{W}_l^{(2)}\boldsymbol{P}_{l-1}^T \rangle_F + \langle \boldsymbol{W}_{l+1}^{(1)}, \boldsymbol{P}_{l+1}\boldsymbol{W}_{l+1}^{(2)}\boldsymbol{P}_l^T \rangle_F$$

$$= \arg\max_{\boldsymbol{P}_l} \langle \boldsymbol{P}_l, \boldsymbol{W}_l^{(1)}\boldsymbol{P}_{l-1}\boldsymbol{W}_l^{(2)^T} + \boldsymbol{W}_{l+1}^{(1)^T}\boldsymbol{P}_{l+1}\boldsymbol{W}_{l+1}^{(2)} \rangle_F$$

$$(7)$$

Consequently, the optimal solution for each iteration can be obtained using LAP-solving techniques (Kuhn, 1955; Jonker & Volgenant, 1986). The whole process of weight matching is presented in Algorithm 1.

---

**Algorithm 1** Permutation Coordinate Descent

> **Initialize:** $\boldsymbol{P}_1 \leftarrow \boldsymbol{I}, ..., \boldsymbol{P}_{L-1} \leftarrow \boldsymbol{I}$
> **repeat**
>   **for** $l \in$ RandomPermutation$(1, ..., L)$ **do**
>     $\boldsymbol{P}_l \leftarrow$ SolveLAP$(\boldsymbol{P}_l, \boldsymbol{W}_l^{(1)}\boldsymbol{P}_{l-1}\boldsymbol{W}_l^{(2)^T} + \boldsymbol{W}_{l+1}^{(1)^T}\boldsymbol{P}_{l+1}\boldsymbol{W}_{l+1}^{(2)})$
>   **end for**
> **until** convergence

---

### A.3 PRUNING PROCESS AND SOME DISCUSSIONS

In our experiments, we use local structure pruning with different pruning rate and have explored three pruning methods, including:

1. **Simple Pruning**: After pruning, the pruned models are directly used for weight matching.

2. **Lottery Ticket Hypothesis**: After pruning the trained model and obtaining mask, the model is retrained from the initial state applying the mask during training. The resulting winning ticket is used for weight matching.

3. **Pruning Followed by Fine-tuning**: After pruning, the model's performance is improved to approximately the pre-pruning level using fine-tuning, and then weight matching is performed for the fine-tuned models.

We found that only the method (3) **Pruning Followed by Fine-tuning** yielded favorable results, and the results are presented in section 5. In contrast, the results of method (1) **Simple Pruning** and (2) **Lottery Ticket Hypothesis** are shown in subsection A.3. Here, we provide possible reasons for the failure of these two methods respectively.

1. **Simple Pruning**: From Figure 7 and Figure 6, it can be observed that after using (1) **Simple Pruning** as the pruning method, the loss barrier between the two models is quite low. However, due to the loss of model accuracy caused by pruning, their accuracy remains poor even after model fusion.

2. **Lottery Ticket Hypothesis**: For **Simple Pruning**, after using local unstructured pruning, each neuron within the same layer still follows nearly the same distribution (described by $Y_{r+1}, ..., Y_n$ in Theorem D.1). Then by Theorem 3.3, the connectivity between neural networks can be maintained. **Pruning Followed by Fine-tuning** method only fine-tune the model obtaining from simple pruning, and the neuron distribution it induces is similar to that caused by simple pruning. While for **Lottery Ticket Hypothesis**, it retrain from the initial state, using different fixed masks, and then the neuron of winning tickets don't follows the same distribution. Then neuron matching is difficult.(Like the distance between $(0, w)$ and $(w, 0)$ is always larger that distance between $(0, w)$ and $(0, w)$ in the average sense, where w is a random variable.). The result is shown in Figure 8.

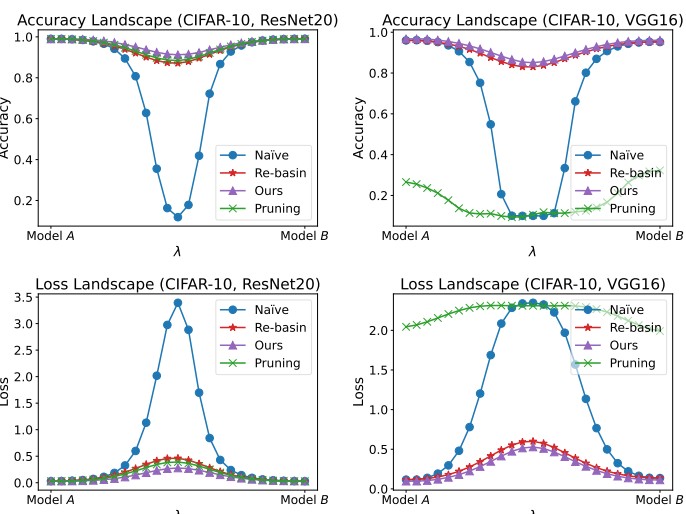

Figure 6: **Train accuracy and loss of Simple Pruning method.**

### A.4 MORE RESULTS OF **PRUNING FOLLOWED BY FINE-TUNING**

We also have more results for **Pruning Followed by Fine-tuning** method on training and test dataset of CIFAR-100 with ResNet20 and CIFAR-10 with VGG16 models (Figure 10 and Figure 11). The corresponding result to Figure 4 on training dataset is in Figure 9

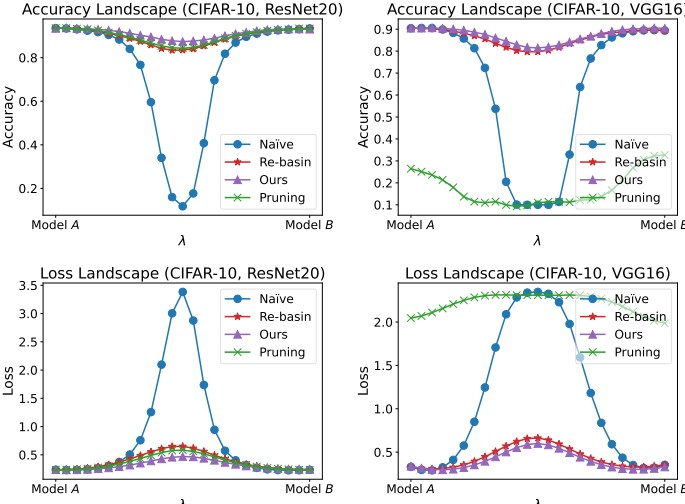

Figure 7: **Test accuracy and loss of Simple Pruning method.**

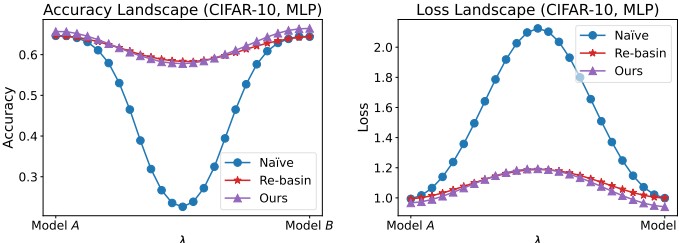

Figure 8: **Accuracy and loss of the Lottery Ticket Hypothesis method**

## B    EXPLANATIONS OF DEFINITION OF NON-UNIFORMITY

There may be 2 doubts regarding neuron distribution entropy:

1. How do we find entropy is related to the linear mode connectivity of neural networks? Why not other more conventional metrics like standard deviation?

2. Why do we employ discrete estimation Shannon entropy instead of the more conventional continuous entropy or discrete entropy?

Explanations for these two doubts are provided below.

**Explanation 1:** For our post-re-basin linear mode connectivity (measure loss barrier after re-basin), our analysis (Theorem 3.1) shows connection of this post-re-basin linear connectivity with Random Bipartite Matching problem. In Random Bipartite Matching problem, the characteristic measure of neuron distribution associated with bounds is entropy(Theorem 3.2).

**Explanation 2:** The reason for using discrete estimation Shannon entropy is that, when measuring the distribution of neurons, especially during pruning, we encounter both discrete random variables with fixed values and continuous random variables. The continuous entropy of discrete random variables is not meaningful(which is $-\infty$), so in this paper, we discretize continuous probability distributions and unify the analysis of the two types of random distributions using discrete entropy. For a detailed discussion, please refer to Chakrabarti et al. (2005); Marsh (2013). Numerically, its effect is to impose a lower bound on the commonly used continuous entropy, while in terms of information theory, it compensates for the part $\log \Delta$ that is omitted during the definition of continuous entropy to give entropy an absolute measure of information. Discrete estimation Shannon entropy for discrete random variables is the discrete entropy itself and discrete estimation Shannon entropy for 1D continuous random variables (high-dimensional continuous random variables are similar) is defined as follows in details (Chakrabarti et al., 2005; Marsh, 2013):

- Given a continuous random variable w and choose an range $\mathbb{L}$ where it is most likely to occur (Which means that $1 - P(\mathbb{L}) < \epsilon$ for some small $\epsilon > 0$). Then split the range $\mathbb{L}$ into $N$ bins $\mathbb{L}_1, ..., \mathbb{L}_N$ equally with $\Delta = \frac{|\mathbb{L}|}{N}$ as the size of each bin.

- Calculate $p_i = \int_{\mathbb{L}_i} p(x)dx$, where $p$ is probability density function of random variable w

- Then the discrete estimation Shannon entropy is $H_\Delta(p) = -\sum_{i=1}^{N} p_i \log p_i$

The relationship between discrete estimation Shannon entropy and continuous entropy for continuous random variable is shown as follow:

$$
\begin{aligned}
H_\Delta(p) &= -\sum_{i=1}^{N} p_i \log p_i \\
&= -\sum_{i=1}^{N} p(x_i)\Delta \log(p(x_i)\Delta) \\
&= -\sum_{i=1}^{N} p(x_i)\log(p(x_i))\Delta + \sum_{i=1}^{N} p(x_i)\Delta \log \Delta \\
&= -\int_{\mathbb{L}} p(x)\log(p(x))dx + \log \Delta \\
&= H(p) + \log \Delta
\end{aligned}
\tag{8}
$$

where $x_i$ is an arbitrary point in the bin $\mathbb{L}_i$.

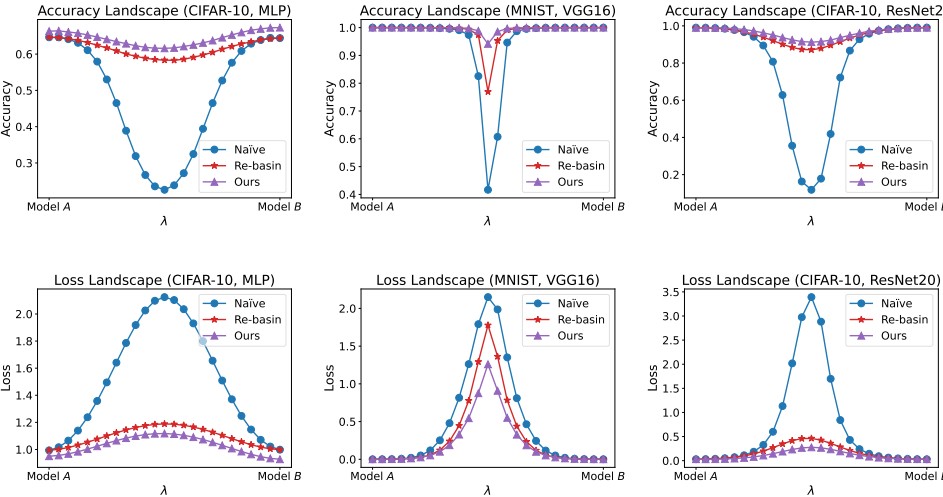

Figure 9: **Trained accuracy and loss corresponding to subsection 4.3.**

## C  OTHER THEORETICAL RESULTS

In the following analysis, we define $D(\boldsymbol{W}_i^{(1)}, \boldsymbol{W}_i^{(2)}) = \min_{\boldsymbol{P} \in \mathbb{S}_{n_i}} ||\boldsymbol{P}\boldsymbol{W}_i^{(1)} - \boldsymbol{W}_i^{(2)}||_2$ as the minimum distance after permutation, $D(\boldsymbol{\theta}_1, \boldsymbol{\theta}_2) = \min_\pi ||\pi(\boldsymbol{\theta}_1) - \boldsymbol{\theta}_2||$ as the minimum distance of all parameter after permutation, $\tilde{D}_E(W_i^{(1)}, W_i^{(2)}) = E_{W_i^{(1)}, W_i^{(2)}} ||W_1 - W_2||_2$ as the expectation of distance of $W_i^{(1)}$ and $W_i^{(2)}$ and the definitions of $W_i^{(1)}, W_i^{(2)}$, and $D_E(W_i^{(1)}, W_i^{(2)})$ are show in section 3.

### C.1  RE-BASIN METHOD DEDUCE LINEAR MODE CONNECTIVITY

Here we extend the result of

**Theorem C.1 (Extension of Theorem 3.1 in Entezari et al. (2021))** *When $b_i \in \sqrt{\frac{1}{n_i + n_{i-1}}}$ (you may refer to to the definition of $b_i$ in Theorem 3.3), and neural network with wide hidden layers $n_i \gg n_{in}, n_{out}$ and $n_i \sim n, i \neq 0, L$, with probability $1 - \delta$*

$$
B_{L-1} = \tilde{O}(n^{-\frac{1}{2}} b_x)
\tag{9}
$$

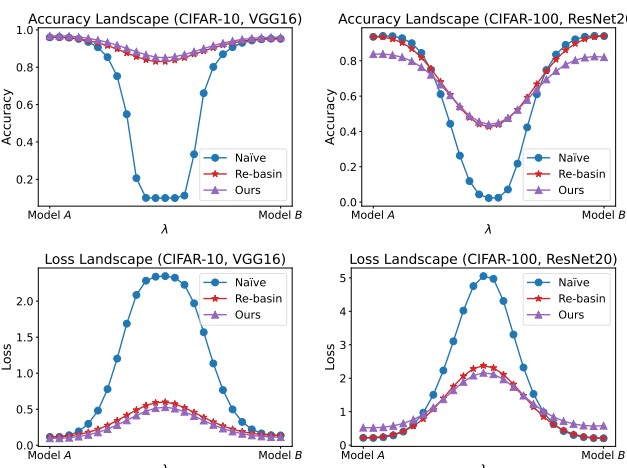

Figure 10: **Trained accuracy and loss of other finetuning experiments.**

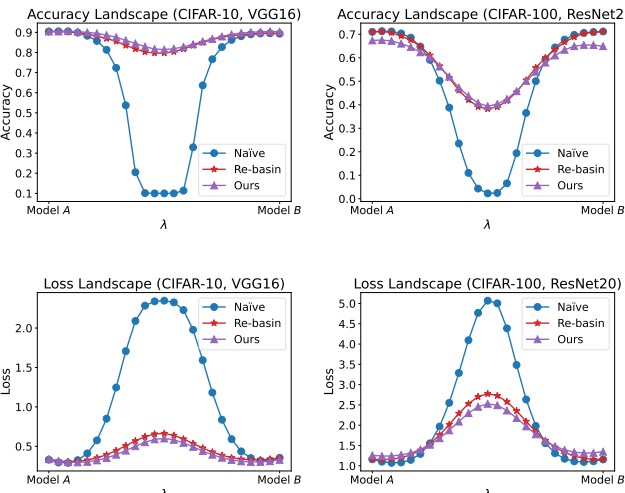

Figure 11: **Tested accuracy and loss of other finetuning experiments.**

*Thus the loss barrier is bounded by $D(\boldsymbol{W}_1, \boldsymbol{W}_2) = \tilde{O}(n^{-\frac{1}{2}} b_x^2)$, which is an extension of the conclusion in Entezari et al. (2021) to multi-layer neural networks, and it further improves the order derived in Entezari et al. (2021).*

*If remove the re-basin operation and the loss barrier is bounded by $\tilde{O}(||\boldsymbol{x}||_2)$, which shows that re-basin deduce linear mode connectivity.*

**Remark** Many commonly used initialization methods satisfy the condition $b_i \in \sqrt{\frac{1}{n_i + n_{i-1}}}$ to ensure that the variance remains unchanged as information passes through each layer (Glorot & Bengio, 2010; He et al., 2015). Therefore, setting this condition is reasonable and does not lead to limitations in the inference.

For proving Theorem C.1, we need the following Lemma.

**Lemma C.2** *Consider matrices $\boldsymbol{W}_1, \boldsymbol{W}_2 \in \mathbb{R}^{n_1 \times n_2}$ whose rows are i.i.d. random vector with uniform distribution in $[0,1]^{n_2}$, if $n_2 > 2$, then with probability $1 - \delta$, there exists a constant $c_\delta$ associated with $\delta$ such that $D(\boldsymbol{W}_1, \boldsymbol{W}_2) \leq c_\delta n_1^{\frac{1}{2} - \frac{2}{n_2}}$. If we do not permute the matrices, then with $1 - \delta$ probability the bound for $||\boldsymbol{W}_1 - \boldsymbol{W}_2||_2$ become $||\boldsymbol{W}_1 - \boldsymbol{W}_2||_2 \leq \tilde{c}_\delta \sqrt{n_1 n_2}$ for a constant $\tilde{c}_\delta$ associated with $\delta$.*

**Proof of Theorem C.1**  By Lemma C.2, $||\boldsymbol{W}_i^{(1)} - \boldsymbol{W}_i^{(2)}||_2 = \tilde{O}(b_i n_i^{\frac{1}{2} - \frac{1}{n_{i-1}}})$, then we have

$$B^{(i+1)} = \tilde{O}(n_{i+1}^{\frac{1}{2}} b_{i+1}(B_i + D_{i-1}))$$
$$D_i = \tilde{O}(n_{i+1}^{\frac{1}{2}} b_{i+1} D_{i-1} + b_i n_i^{\frac{1}{2} - \frac{1}{n_{i-1}}} \Pi_{j=1}^i n_j^{\frac{1}{2}} b_j ||\boldsymbol{x}||_2) \qquad (10)$$

Under the assumption $b_i \sim \sqrt{\frac{1}{n_i + n_{i-1}}}$, for neural network with wide hidden layers, we have:

$$D_i = \tilde{O}(D_{i-1} + n_i^{-\frac{1}{n_{i-1}}} ||\boldsymbol{x}||_2)$$
$$D_0 = \tilde{O}(n_1^{-\frac{1}{n_0}} ||\boldsymbol{x}||_2) \Rightarrow \qquad (11)$$
$$D_i = \tilde{O}(||\boldsymbol{x}||_2)$$

For $B_i$ we have

$$B^{L-1} = \tilde{O}(n_{L-1}^{-\frac{1}{2}}(B_{L-2} + D_{L-3}))$$
$$B^{(i+1)} = \tilde{O}((B_i + D_{i-1})), i < L - 2 \qquad (12)$$

Then we have the bound for the neural network with wide hidden layers:

$$B_{L-1} = \tilde{O}(n^{-\frac{1}{2}} ||\boldsymbol{x}||_2) \qquad (13)$$

If we remove the re-basin process and the matrices are not permuted, the recursive structure is

$$B^{(i+1)} = \tilde{O}(n_{i+1}^{\frac{1}{2}} b_{i+1}(B_i + D_{i-1}))$$
$$D_i = \tilde{O}(n_{i+1}^{\frac{1}{2}} b_{i+1} D_{i-1} + b_i \sqrt{n_i n_{i-1}} \Pi_{j=1}^i n_j^{\frac{1}{2}} b_j ||\boldsymbol{x}||_2) \qquad (14)$$

And repeat the process above, we will have $B_{L-1} = \tilde{O}(||\boldsymbol{x}||_2)$

**Proof for Lemma C.2**  Here we write the $j$th row of matrix $\boldsymbol{W}_i$ as $\boldsymbol{w}_{j,:}^{(i)}$ and $E_{\boldsymbol{W}_1, \boldsymbol{W}_2}(\min_{\boldsymbol{P} \in \mathbb{S}_{n_1}} ||\boldsymbol{P}\boldsymbol{W}_1 - \boldsymbol{W}_2||_2^2) = E(\min_\pi \sum_{i=1}^{n_1} ||\boldsymbol{w}_{i,:}^{(1)} - \boldsymbol{w}_{i,:}^{(2)}||_2^2)$, which is the formula of the random Euclidean matching problem Benedetto & Caglioti (2020). Here $\pi$ is the permutation of set $\{1, ..., n_1\}$. Define $Z_{n_1} = n_1^{-\frac{1}{2} + \frac{1}{n_2}} \min_\pi (\sum_{i=1}^{n_1} ||\boldsymbol{w}_{i,:}^{(1)} - \boldsymbol{w}_{\pi(i),:}^{(2)}||_2^2)^{\frac{1}{2}}$ (Boutet de Monvel & Martin (2002), while they represent $Z_n$ in the form of the Wasserstein distance.) The Theorem 1.1 in Boutet de Monvel & Martin (2002) proves that

$$\lim_{n_1 \to \infty} E(Z_{n_1}^2) = c, \lim_{n_1 \to \infty} E(Z_{n_1}) = c^{\frac{1}{2}} \qquad (15)$$

where $c$ is a constant. Then ignoring lower-order small terms, it holds that for large $n_1$

$$E(Z_{n_1}^2) \approx c, E(Z_{n_1}) \approx c^{\frac{1}{2}} \tag{16}$$

For $r > 0$, the following concentration inequality holds

$$P(|Z_{n_1} - E[Z_{n_1}]| \geq r) \leq 2\exp(-cr^2 n_1^{1-\frac{2}{n_2}}) \tag{17}$$

Then for $D(\boldsymbol{W}_1, \boldsymbol{W}_2) = \min_{\boldsymbol{P} \in \mathbb{S}_{n_1}} ||\boldsymbol{P}\boldsymbol{W}_1 - \boldsymbol{W}_2||_2$, we have inequality

$$P(|n_1^{-\frac{1}{2}+\frac{1}{n_2}} D(\boldsymbol{W}_1, \boldsymbol{W}_2) - E[Z_{n_1}]| \geq r) \leq 2\exp(-c_2 r^2 n_1^{1-\frac{2}{n_2}}) \tag{18}$$

Thus with probability $1 - \delta$, we have

$$n_1^{-\frac{1}{2}+\frac{1}{n_2}} D(\boldsymbol{W}_1, \boldsymbol{W}_2) \leq \frac{n_1^{\frac{1}{n_2}-\frac{1}{2}}}{\sqrt{c_2}} \ln^{\frac{1}{2}} \frac{2}{\delta} + c_1^{\frac{1}{2}} \tag{19}$$

Then we have the bound for $D(\boldsymbol{W}_1, \boldsymbol{W}_2)$ as

$$D(\boldsymbol{W}_1, \boldsymbol{W}_2) \leq c_\delta n_1^{\frac{1}{2}-\frac{1}{n_2}} \tag{20}$$

for some constant $c_\delta$ associated with $\delta$. Then we calculate the bound of $||\boldsymbol{W}_1 - \boldsymbol{W}_2||_2$. For $\boldsymbol{w}_{j,:}^{(1)}$, $\boldsymbol{w}_{j,:}^{(2)}$ are i.i.d, then $E(||\boldsymbol{W}_1 - \boldsymbol{W}_2||_2^2) = n_1 E(||\boldsymbol{w}_{0,:}^{(1)} - \boldsymbol{w}_{0,:}^{(2)}||_2^2)$, and $E(||\boldsymbol{w}_{0,:}^{(1)} - \boldsymbol{w}_{0,:}^{(2)}||_2^2)$ can be calculated as

$$\begin{aligned} E(||\boldsymbol{w}_{0,:}^{(1)} - \boldsymbol{w}_{0,:}^{(2)}||_2^2) &= E(\sum_{j=1}^{n_2} (\boldsymbol{w}_{0,j}^{(1)} - \boldsymbol{w}_{0,j}^{(2)})^2) \\ &= \sum_{j=1}^{n_2} E((\boldsymbol{w}_{0,j}^{(1)} - \boldsymbol{w}_{0,j}^{(2)})^2) \\ &= c_3 n_2 \end{aligned} \tag{21}$$

Where $c_3 = E_{\mathrm{x,y} \sim U([0,1]), i.i.d}((\mathrm{x} - \mathrm{y})^2)$ is constant. Thus $E(||\boldsymbol{W}_1 - \boldsymbol{W}_2||_2^2) = c_3 n_1 n_2$. By the concentration inequality, we have

$$P(|||\boldsymbol{W}_1 - \boldsymbol{W}_2||_2^2 - E(||\boldsymbol{W}_1 - \boldsymbol{W}_2||_2^2)| \geq t) \leq 2e^{\frac{-2t^2}{n_1 n_2}} \tag{22}$$

Then with probability $1 - \delta$, we have

$$\begin{aligned} ||\boldsymbol{W}_1 - \boldsymbol{W}_2||_2^2 &\leq E(||\boldsymbol{W}_1 - \boldsymbol{W}_2||_2^2) + \sqrt{\frac{n_1 n_2}{2} \ln \frac{2}{\delta}} \\ &\leq c_3 n_1 n_2 + \sqrt{\frac{n_1 n_2}{2} \ln \frac{2}{\delta}} \end{aligned} \tag{23}$$

So there exist constant $\tilde{c}_\delta$ associated with $\delta$ such that $||\boldsymbol{W}_1 - \boldsymbol{W}_2||_2 \leq \tilde{c}_\delta \sqrt{n_1 n_2}$, with probability $1 - \delta$.

## D  MISSING PROOFS

Some definition of symbols can be found in Appendix C.

### D.1  MISSING PROOF FOR THEOREM 3.1

**Theorem 3.1**  If each row $\boldsymbol{w}_{j,:}^{(i)}$ of the weight matrix $\boldsymbol{W}_i$ of layer $i$ follows distribution $\mathbb{R}^{n_{i-1}} \ni \mathbf{w} = (\mathrm{w}_1, ..., \mathrm{w}_{n_{i-1}}) \sim P$ i.i.d. with $b_i \triangleq \sqrt{\sum_{j=1}^{n_{i-1}} Var(\mathrm{w}_j)}$, and the input of the neural network $x$ is bounded $||x||_2 < b_x$, then for any $\delta > 0$, with probability $1 - \delta$,

$$\sup_{\alpha \in [0,1]} |f_{\alpha\boldsymbol{\theta}_1 + (1-\alpha)\boldsymbol{\theta}_2}(\boldsymbol{x}) - \alpha f_{\boldsymbol{\theta}_1}(\boldsymbol{x}) - (1-\alpha) f_{\theta_2}(\boldsymbol{x})| \leq B_{L-1} b_x \tag{24}$$

And $B_{L-1}$ is bounded by the following recursive equations

$$\begin{aligned} B_{i+1} &= \tilde{O}(n_{i+1}^{\frac{1}{2}} b_{i+1}(B_i + D_{i-1})) \\ D_i &= \tilde{O}(n_{i+1}^{\frac{1}{2}} b_{i+1} D_{i-1} + D_E(\boldsymbol{W}_{i+1}^{(1)}, \boldsymbol{W}_{i+1}^{(2)}) \Pi_{j=1}^i n_j^{\frac{1}{2}} b_j b_x) \\ B_1 &= 0, D_0 = \tilde{O}(D_E(\boldsymbol{W}_1^{(1)}, \boldsymbol{W}_1^{(2)}) b_x) \end{aligned} \tag{25}$$

where $D_E(\boldsymbol{W}_i^{(1)}, \boldsymbol{W}_i^{(2)}) = E_{\boldsymbol{W}_i^{(1)}, \boldsymbol{W}_i^{(2)}} \min_{\boldsymbol{P} \in \mathbb{S}_{n_i}} ||\boldsymbol{P}\boldsymbol{W}_1 - \boldsymbol{W}_2||_2$ as the expectation of minimum distance after permutation and $\mathbb{S}_{n_i}$ is the set of permutation matrices with size $n_i \times n_i$.

**Proof** For $L$-layer fully connected neural network $f_{\boldsymbol{\theta}}(x)$ with parameter $\boldsymbol{\theta} = (\boldsymbol{W}_1, ..., \boldsymbol{W}_L)$, we define the function represented by the first $i$ layer as $f^{(i)}_{\boldsymbol{W}_j|_{j=1}^i}(x) = f^{(i)}_{\boldsymbol{W}_i}(...f^{(1)}_{\boldsymbol{W}_1}(\boldsymbol{x}))$ and for two neural networks with parameters $\boldsymbol{\theta}_1 = (\boldsymbol{W}_1^{(1)}, ..., )$ and $\boldsymbol{\theta}_2 = (\boldsymbol{W}_1^{(1)}, ..., )$ we define the first $i+1$ layer difference (with parameter $\alpha$)as

$$
\begin{aligned}
B^{(i+1)} &= \sup_\alpha B_\alpha^{(i+1)} \\
&= \sup_\alpha ||(\alpha \boldsymbol{W}_{i+1}^{(1)} + (1-\alpha)\boldsymbol{W}_{i+1}^{(2)})f^{(i)}_{\alpha \boldsymbol{W}_j^{(1)}+(1-\alpha)\boldsymbol{W}_j^{(2)}|_{j=1}^i}(\boldsymbol{x}) - \\
&\quad \alpha \boldsymbol{W}_{i+1}^{(1)} f^{(i)}_{\boldsymbol{W}_j^{(1)}|_{j=1}^i}(\boldsymbol{x}) - (1-\alpha)\boldsymbol{W}_{i+1}^{(2)} f^{(i)}_{\boldsymbol{W}_j^{(2)}|_{j=1}^i}(\boldsymbol{x})||_2
\end{aligned}
\tag{26}
$$

Then we begin to calculate the bound $B_\alpha^{(i+1)}$ as

$$
\begin{aligned}
B_\alpha^{(i+1)} &= ||(\alpha \boldsymbol{W}_{i+1}^{(1)} + (1-\alpha)\boldsymbol{W}_{i+1}^{(2)})\sigma((\alpha \boldsymbol{W}_j^{(1)} + (1-\alpha)\boldsymbol{W}_j^{(2)})f^{(i-1)}_{\alpha \boldsymbol{W}_j^{(1)}+(1-\alpha)\boldsymbol{W}_j^{(2)}|_{j=1}^{i-1}}(\boldsymbol{x})) - \\
&\quad \alpha \boldsymbol{W}_{i+1}^{(1)}\sigma(\boldsymbol{W}_i^{(1)} f^{(i-1)}_{\boldsymbol{W}_j^{(1)}|_{j=1}^{i-1}}(\boldsymbol{x})) - (1-\alpha)\boldsymbol{W}_{i+1}^{(2)}\sigma(\boldsymbol{W}_i^{(2)} f^{(i-1)}_{\boldsymbol{W}_j^{(2)}|_{j=1}^{i-1}}(\boldsymbol{x}))||_2 \\
&\leq ||\alpha \boldsymbol{W}_{i+1}^{(1)}||_2 ||\sigma((\alpha \boldsymbol{W}_i^{(1)} + (1-\alpha)\boldsymbol{W}_i^{(2)})f^{(i-1)}_{\alpha \boldsymbol{W}_j^{(1)}+(1-\alpha)\boldsymbol{W}_j^{(2)}|_{j=1}^{i-1}}(\boldsymbol{x})) - \\
&\quad\quad\quad\quad\quad\quad\quad\quad\quad\quad\quad\quad\quad\quad\quad\quad \sigma(\boldsymbol{W}_i^{(1)} f^{(i-1)}_{\boldsymbol{W}_j^{(1)}|_{j=1}^{i-1}}(\boldsymbol{x}))||_2 + \\
&\quad ||(1-\alpha)\boldsymbol{W}_{i+1}^{(2)}||_2 ||\sigma((\alpha \boldsymbol{W}_i^{(1)} + (1-\alpha)\boldsymbol{W}_i^{(2)})f^{(i-1)}_{\alpha \boldsymbol{W}_j^{(1)}+(1-\alpha)\boldsymbol{W}_j^{(2)}|_{j=1}^{i-1}}(\boldsymbol{x})) - \\
&\quad\quad\quad\quad\quad\quad\quad\quad\quad\quad\quad\quad\quad\quad\quad\quad \sigma(\boldsymbol{W}_i^{(2)} f^{(i-1)}_{\boldsymbol{W}_j^{(2)}|_{j=1}^{i-1}}(\boldsymbol{x}))||_2
\end{aligned}
\tag{27}
$$

The Hoeffding's inequality tells that for a random vector $\mathbf{v} \in \mathbb{R}^d$ and if elements of $\mathbf{v}$ are independent and with 0 mean value and bound $[-b_1, b_1], ..., [-b_d, b_d]$ respectively, then for a coefficient vector $\boldsymbol{r} \in \mathbb{R}^d$ it holds

$$
P(|\mathbf{v}\boldsymbol{r}| \geq t) \leq 2e^{\frac{-dt^2}{2||\boldsymbol{r}||_2^2 \sum_{i=1}^d b_i^2}}
\tag{28}
$$

Then with probability $1 - k\delta$, we have

$$
|\mathbf{v}\boldsymbol{r}| < ||\boldsymbol{r}||_2 \sqrt{\frac{2\sum_{i=1}^d b_i^2 ln 2k/\delta}{d}}
\tag{29}
$$

Apply the inequation above for random matrix $\mathbf{W}_i \in \mathbb{R}^{n_i \times n_{i-1}}$, with initialization that $\frac{\sum_{j=1}^d b_{j,i}^2}{n_{i-1}} = b_i^2$, with probability 1-k$\delta$, we have(here using inequation above for $n_i$ times and using the union bound):

$$
||\boldsymbol{W}_i \boldsymbol{r}||_2 < ||\boldsymbol{r}||_2 \sqrt{2n_i b_i^2 ln \frac{2kn_i}{\delta}}
\tag{30}
$$

Then calculate $B_\alpha^{(i+1)}$, with the Lipschitz assumption of activate function $\sigma$

$$
\begin{aligned}
B_\alpha^{(i+1)} &\leq \sqrt{2n_{i+1} b_{i+1}^2 ln \frac{2kn_{i+1}}{\delta}} \{\alpha ||(\alpha \boldsymbol{W}_i^{(1)} + (1-\alpha)\boldsymbol{W}_i^{(2)})f^{(i-1)}_{\alpha \boldsymbol{W}_j^{(1)}+(1-\alpha)\boldsymbol{W}_j^{(2)}|_{j=1}^{i-1}}(\boldsymbol{x}) - \\
&\quad\quad\quad\quad\quad\quad\quad\quad\quad\quad \boldsymbol{W}_i^{(1)} f^{(i-1)}_{\boldsymbol{W}_j^{(1)}|_{j=1}^{i-1}}(\boldsymbol{x})||_2 + \\
&\quad (1-\alpha)(\alpha \boldsymbol{W}_i^{(1)} + (1-\alpha)\boldsymbol{W}_i^{(2)})f^{(i-1)}_{\alpha \boldsymbol{W}_j^{(1)}+(1-\alpha)\boldsymbol{W}_j^{(2)}|_{j=1}^{i-1}}(\boldsymbol{x}) - \boldsymbol{W}_i^{(2)} f^{(i-1)}_{\boldsymbol{W}_j^{(2)}|_{j=1}^{i-1}}(\boldsymbol{x})||_2\}
\end{aligned}
\tag{31}
$$

We first compute the first term on the right-hand side of the inequality above

$$
\begin{aligned}
&||(\alpha \boldsymbol{W}_i^{(1)} + (1-\alpha)\boldsymbol{W}_i^{(2)}) f_{\alpha \boldsymbol{W}_j^{(1)} + (1-\alpha)\boldsymbol{W}_j^{(2)}|_{j=1}^{i-1}}^{(i-1)}(\boldsymbol{x}) - \boldsymbol{W}_i^{(1)} f_{\boldsymbol{W}_j^{(1)}|_{j=1}^{i-1}}^{(i-1)}(\boldsymbol{x})||_2 \\
&\leq ||(\alpha \boldsymbol{W}_i^{(1)} + (1-\alpha)\boldsymbol{W}_i^{(2)}) f_{\alpha \boldsymbol{W}_j^{(1)} + (1-\alpha)\boldsymbol{W}_j^{(2)}|_{j=1}^{i-1}}^{(i-1)}(\boldsymbol{x}) \\
&\qquad\qquad - \alpha \boldsymbol{W}_i^{(1)} f_{\boldsymbol{W}_j^{(1)}|_{j=1}^{i-1}}^{(i-1)}(\boldsymbol{x}) - (1-\alpha)\boldsymbol{W}_i^{(2)} f_{\boldsymbol{W}_j^{(2)}|_{j=1}^{i-1}}^{(i-1)}(\boldsymbol{x})||_2 + \\
&(1-\alpha)||\boldsymbol{W}_i^{(1)} f_{\boldsymbol{W}_j^{(1)}|_{j=1}^{i-1}}^{(i-1)}(\boldsymbol{x}) - \boldsymbol{W}_i^{(2)} f_{\boldsymbol{W}_j^{(2)}|_{j=1}^{i-1}}^{(i-1)}(\boldsymbol{x})||_2 \\
&= B_i + (1-\alpha)D_{i-1}
\end{aligned}
\tag{32}
$$

Here we define $D_i = ||\boldsymbol{W}_{i+1}^{(1)} f_{\boldsymbol{W}_j^{(1)}|_{j=1}^{i}}^{(i)}(x) - \boldsymbol{W}_{i+1}^{(2)} f_{\boldsymbol{W}_j^{(2)}|_{j=1}^{i}}^{(i)}(x)||_2$, and $D_i$ have the following inequality

$$
\begin{aligned}
D_i &= ||\boldsymbol{W}_{i+1}^{(1)} f_{\boldsymbol{W}_j^{(1)}|_{j=1}^{i}}^{(i)}(x) - \boldsymbol{W}_{i+1}^{(2)} f_{\boldsymbol{W}_j^{(2)}|_{j=1}^{i}}^{(i)}(\boldsymbol{x})||_2 \\
&\leq ||\boldsymbol{W}_{i+1}^{(1)} f_{\boldsymbol{W}_j^{(1)}|_{j=1}^{i}}^{(i)}(\boldsymbol{x}) - \boldsymbol{W}_{i+1}^{(2)} f_{\boldsymbol{W}_j^{(1)}|_{j=1}^{i}}^{(i)}(\boldsymbol{x})|| + ||\boldsymbol{W}_{i+1}^{(2)} f_{\boldsymbol{W}_j^{(1)}|_{j=1}^{i}}^{(i)}(\boldsymbol{x}) - \boldsymbol{W}_{i+1}^{(2)} f_{\boldsymbol{W}_j^{(2)}|_{j=1}^{i}}^{(i)}(\boldsymbol{x})||_2 \\
&\leq ||(\boldsymbol{W}_{i+1}^{(1)} - \boldsymbol{W}_{i+1}^{(2)}) f_{\boldsymbol{W}_j^{(1)}|_{j=1}^{i}}^{(i)}(\boldsymbol{x})||_2 + ||\boldsymbol{W}_{i+1}^{(2)}(f_{\boldsymbol{W}_j^{(1)}|_{j=1}^{i}}^{(i)}(\boldsymbol{x}) - f_{\boldsymbol{W}_j^{(2)}|_{j=1}^{i}}^{(i)}(\boldsymbol{x}))||_2 \\
&\leq ||(\boldsymbol{W}_{i+1}^{(1)} - \boldsymbol{W}_{i+1}^{(2)}) f_{\boldsymbol{W}_j^{(1)}|_{j=1}^{i}}^{(i)}(\boldsymbol{x})||_2 + ||f_{\boldsymbol{W}_j^{(1)}|_{j=1}^{i}}^{(i)}(\boldsymbol{x}) - f_{\boldsymbol{W}_j^{(2)}|_{j=1}^{i}}^{(i)}(\boldsymbol{x}))||_2 \\
&\leq \sqrt{2n_{i+1}b_{i+1}^2 ln\frac{2kn_{i+1}}{\delta}} D_{i-1} + ||(\boldsymbol{W}_{i+1}^{(1)} - \boldsymbol{W}_{i+1}^{(2)}) f_{\boldsymbol{W}_j^{(1)}|_{j=1}^{i}}^{(i)}(\boldsymbol{x})||_2
\end{aligned}
\tag{33}
$$

For $||f_{\boldsymbol{W}_j^{(1)}|_{j=1}^{i}}^{(i)}(\boldsymbol{x})||_2 = ||\sigma(\boldsymbol{W}_i^{(1)} f_{\boldsymbol{W}_j^{(1)}|_{j=1}^{i-1}}^{(i-1)}(\boldsymbol{x}))||_2 \leq ||\boldsymbol{W}_i^{(1)} f_{\boldsymbol{W}_j^{(1)}|_{j=1}^{i-1}}^{(i-1)}(\boldsymbol{x})||_2 = \sqrt{2n_i b_i^2 \ln\frac{2kn_i}{\delta}} ||f_{\boldsymbol{W}_j^{(1)}|_{j=1}^{i-1}}^{(i-1)}(\boldsymbol{x})||_2$, then we have

$$
||f_{\boldsymbol{W}_j^{(1)}|_{j=1}^{i}}^{(i)}(\boldsymbol{x})||_2 \leq \Pi_{j=1}^{i} \sqrt{2n_j b_j^2 \ln\frac{2kn_j}{\delta}} ||\boldsymbol{x}||_2
\tag{34}
$$

After summarizing the discussion above, and ignore the lower-order term $\log n_i$(in subsequent analysis, it can indeed be found that $\log n_i$ is a lower-order term).

$$
\begin{aligned}
B^{(i+1)} &\leq \sqrt{2n_{i+1}b_{i+1}^2 \ln\frac{2kn_{i+1}}{\delta}}(B_i + 2\alpha(1-\alpha)D_{i-1}) = \tilde{O}(n_{i+1}^{\frac{1}{2}}b_{i+1}(B_i + D_{i-1})) \\
D_i &= \tilde{O}(n_{i+1}^{\frac{1}{2}}b_{i+1}D_{i-1} + \Pi_{j=1}^{i}n_j^{\frac{1}{2}}b_j||x||_2||W_{i+1}^{(1)} - W_{i+1}^{(2)}||_2)
\end{aligned}
\tag{35}
$$

with initial value $D_0 = \tilde{O}(||W_1^{(1)} - W_1^{(2)}||_2||x||_2)$ and $B_1 = 0$. The discussion above have not consider the optimal permutation, and add the optimal permutation $\pi$, the recursive structure for bound can be expressed as:

$$
\begin{aligned}
B^{(i+1)} &\leq \sqrt{2n_{i+1}b_{i+1}^2 \ln\frac{2kn_{i+1}}{\delta}}(B_i + 2\alpha(1-\alpha)D_{i-1}) = \tilde{O}(n_{i+1}^{\frac{1}{2}}b_{i+1}(B_i + D_{i-1})) \\
D_i &= \tilde{O}(n_{i+1}^{\frac{1}{2}}b_{i+1}D_{i-1} + D_E(W_{i+1}^{(1)}, W_{i+1}^{(2)})\Pi_{j=1}^{i}n_j^{\frac{1}{2}}b_j b_x)
\end{aligned}
\tag{36}
$$

### D.2 MISSING PROOF FOR LEMMA 3.2

**Lemma 3.2** Consider matrices $\boldsymbol{W}_1, \boldsymbol{W}_2 \in \mathbb{R}^{n_1 \times n_2}$ whose rows are i.i.d. random vector of arbitrary Hölder continuous probability density $\rho$ on a bounded open set $\Omega$, for large $n_2 > 2$, the bound for $D(\boldsymbol{W}_1, \boldsymbol{W}_2)$ becomes $D(\boldsymbol{W}_1, \boldsymbol{W}_2) \leq c_\delta n_1^{\frac{1}{2} - \frac{2}{n_2}} e^{c\tilde{H}(\rho)}$ with probability $1 - \delta$ and constant $c$.

**Proof for Lemma 3.2** Before reading this proof, we recommend readers reading Lemma C.2 and its proof first. By Benedetto & Caglioti (2020); Ambrosio et al. (2022); Goldman & Trevisan (2022),

it shows that equation Equation 15 holds even when vectors $\boldsymbol{w}_{i,:}^{(1)}, \boldsymbol{w}_{i,:}^{(2)}$ are not uniformly distributed but rather distributed in a bounded open set $\Omega$ with a Hölder continuous probability density $\rho$, and the limit value is related to the non-uniformity of distribution $\rho$:

$$\lim_{n_1 \to \infty} \sup Z_{n_1}^2 \le c \int_\Omega \rho^{1-\frac{2}{n_2}} \tag{37}$$

which is P-a.s. holds. Combined with the discussion in Lemma C.2, it holds that $D(\boldsymbol{W}_1, \boldsymbol{W}_2) \le c_\delta \sqrt{\int_\Omega \rho^{1-\frac{2}{n_2}}} n_1^{\frac{1}{2}-\frac{2}{n_2}}$. Note that for any continuous distribution $\rho$, the Rényi entropy is defined as $R_\alpha(\rho) = \frac{1}{1-\alpha} ln(\int_\Omega \rho^\alpha)$ and when $\alpha \to 1$, Rényi entropy converges to Shannon entropy $H(\rho)$. Thus the bound of $D(\boldsymbol{W}_1, \boldsymbol{W}_2)$ can be represented as $D(\boldsymbol{W}_1, \boldsymbol{W}_2) \le c_\delta n_1^{\frac{1}{2}-\frac{2}{n_2}} e^{\frac{2}{n_2} R_\alpha(\rho)}$, where $\alpha = 1 - \frac{2}{n_2}$ and when $n_2$ is large, $D(\boldsymbol{W}_1, \boldsymbol{W}_2) \le c_\delta n_1^{\frac{1}{2}-\frac{2}{n_2}} e^{\frac{1}{n_2} H(\rho)} = c_\delta n_1^{\frac{1}{2}-\frac{2}{n_2}} e^{\tilde{H}(\rho)}$ holds.

### D.3 PROOF OF LEMMA 4.1

**Lemma D.1** *Let $x_1, x_2, ..., x_n$ be $n$ i.i.d. random variables with continuous cumulative distribution function $F(x)$ and probability density function $f(x)$. Let $x_{i:n}$ denotes the $i$th order statistics of all the $n$ random variables and $y_{r+1}, ..., y_n$ be the order statistics $x_{r+1:n}, ..., x_{n:n}$ without order. Then the entropy $H_\Delta(0, 0, ..., 0, y_{r+1}, ..., y_n)$ is less than $H_\Delta(x_1, ..., x_n)$, where $H_\Delta(0, 0, ..., 0, y_{r+1}, ..., y_n)$ denotes the approximation discrete entropy of the union distribution of random variables $y_{r+1}, ..., y_n$ and $r$ zero-valued deterministic random variables and $H_\Delta(x_1, ..., x_n)$ similarly.*

**Proof** We first calculate the continuous entropy $H(y_{r+1}, ..., y_n)$ and $H(x_1, ..., x_n)$ and use the relation between continuous entropy $H$ and approximated discrete entropy $H_\Delta$ to calculate the approximated discrete $H_\Delta(0, 0, ..., y_{r+1}, ..., y_n)$ and $H_\Delta(x_1, ..., x_n)$.

Let $x'_i = -x_i$ with cumulative distribution function $F'(x) = 1 - F(-x)$ and probability density function $f'(x) = f(-x)$ and $x'_{i:n}$ denotes the $i$th order statistics of $x'_i$. Then the joint entropy of $x_{r+1:n}, ..., x_{n:n}$ is equal to the joint entropy of the first $s = n - r$ order statistics of $x'_i$, which is denoted as $H'_{1...s;n}$ for simplicity. Park (2005) gives the formula of $H'_{1...s;n}$

$$H'_{1...s;n} = s - log(\frac{n!}{s!}) - n \int_{-\infty}^{\infty} (1 - F'_{s:n-1}(x)) f'(x) log h'(x) dx \tag{38}$$

where $F'_{s:n-1}(x)$ denotes the cumulative distribution function of the $r$th order statics among $n-1$ random variable and $h'(x)$ is the hazard function defined as $h'(x) = f'(x)/(1 - F'(x))$. After adding entropy decreases from the order information (Wong & Chen, 1990), $H(y_{r+1}, ..., y_n) = \log s! + H'_{1...s;n}$. We expand $H(y_{r+1}, ..., y_n)$ as

$$
\begin{aligned}
&H(y_{r+1}, ..., y_n) \\
&= H'_{1...s;n} + \log s! \\
&= s - \log(n!) - n \int_{-\infty}^{\infty} (1 - F'_{s:n-1}(x)) f'(x) \log h'(x) dx \\
&\overset{h'(x)=\frac{f'(x)}{1-F'(x)}=\frac{f(-x)}{F(-x)}}{=} s - \log(n!) - n \int_{-\infty}^{\infty} (1 - F'_{s:n-1}(x)) f'(x) \log \frac{f(-x)}{F(-x)} dx \\
&\overset{F'_{s:n-1}=\sum_{i=s}^{n-1} \binom{n-1}{i} F'(x)^i (1-F'(x))^{n-1-i}}{=} s - \log(n!) - \\
&\qquad n \int_{-\infty}^{\infty} (1 - \sum_{i=s}^{n-1} \binom{n-1}{i}(1 - F(-x))^i F(-x)^{n-1-i}) f'(x) \log \frac{f(-x)}{F(-x)} dx \\
&= s - \log(n!) - n \int_{-\infty}^{\infty} (\sum_{i=0}^{s-1} \binom{n-1}{i}(1 - F(x))^i F(x)^{n-1-i}) f(x) \log \frac{f(x)}{F(x)} dx
\end{aligned}
\tag{39}
$$

Similarly, $H(x_1, ..., x_n) = H'_{1...n;n} + n!$ can be expanded as

$$H(x_1, ..., x_n) = n - \log(n!) - n \int_{-\infty}^{\infty} (\sum_{i=0}^{n-1} \binom{n-1}{i}(1 - F(x))^i F(x)^{n-1-i}) f(x) \log \frac{f(x)}{F(x)} dx \tag{40}$$

So the difference between $H(\mathrm{y}_{r+1}, ..., \mathrm{y}_n)$ and $H(\mathrm{x}_1, ..., \mathrm{x}_n)$ is

$$H(\mathrm{y}_{r+1}, ..., \mathrm{y}_n) - H(\mathrm{x}_1, ..., \mathrm{x}_n) = -r+$$

$$n \int_{-\infty}^{\infty} \left( \sum_{i=n-r}^{n-1} \binom{n-1}{i} (1 - F(x))^i F(x)^{n-1-i} \right) f(x) \log \frac{f(x)}{F(x)} dx$$

$$(41)$$

While $H_\Delta(0, 0, ..., \mathrm{y}_{r+1}, ..., \mathrm{y}_n) = H_\Delta(\mathrm{y}_{r+1}, ..., \mathrm{y}_n) = (n - r)N + H(\mathrm{y}_{r+1}, ..., \mathrm{y}_n)$ and $H_\Delta(\mathrm{x}_1, ..., \mathrm{x}_n) = nN + H(\mathrm{x}_1, ..., \mathrm{x}_n)$, difference between $H_\Delta(0, 0, ..., \mathrm{y}_{r+1}, ..., \mathrm{y}_n)$ and $H_\Delta(\mathrm{x}_1, ..., \mathrm{x}_n)$ is

$$H_\Delta(0, 0, ..., \mathrm{y}_{r+1}, ..., \mathrm{y}_n) - H_\Delta(\mathrm{x}_1, ..., \mathrm{x}_n)$$

$$= -rN - r + n \int_{-\infty}^{\infty} \left( \sum_{i=n-r}^{n-1} \binom{n-1}{i} (1 - F(x))^i F(x)^{n-1-i} \right) f(x) \log \frac{f(x)}{F(x)} dx \qquad (42)$$

Because we are discussing the discrete estimated entropy $H_\Delta$, according to the definition of $H_\Delta$

$$H_\Delta(0, 0, ..., \mathrm{y}_{r+1}, ..., \mathrm{y}_n) - H_\Delta(\mathrm{x}_1, ..., \mathrm{x}_n)$$

$$= r \log \Delta - r + n \sum_{i=n-r}^{n-1} \sum_{k=1}^{N} \binom{n-1}{i} (1 - F(x_k^*))^i F(x_k^*)^{n-1-i} f(x_k^*) \log \frac{f(x_k^*)}{F(x_k^*)} \Delta$$

$$= -r \log \Delta - r + n \sum_{i=n-r}^{n-1} \sum_{k=1}^{N} \binom{n-1}{i} (1 - F(x_k^*))^i F(x_k^*)^{n-1-i} p_k \log \frac{p_k}{\sum_{j=1}^{k} p_k}$$

$$- n \sum_{i=n-r}^{n-1} \sum_{k=1}^{N} \binom{n-1}{i} (1 - F(x_k^*))^i F(x_k^*)^{n-1-i} p_k \log \Delta \qquad (43)$$

$$\overset{\log \frac{p_k}{\sum_{j=1}^{k} p_k} < 0}{\leq} -r \log \Delta - r - n \left[ \log \Delta \sum_{i=n-r}^{n-1} \int_{-\infty}^{\infty} \binom{n-1}{i} (1 - F(x))^i F(x)^{n-1-i} dF(x) \right]$$

$$\overset{\int_0^1 (1-r)^i r^{n-1-i} dr = \frac{1}{n} \binom{n-1}{i}}{=} -r < 0$$

where $N$ is the number of bins for discretion and $(x_1^*, ..., x_N^*)$ are $N$ arbitrary points in the $N$ bins respectively.

