# OpenReview forum: "Unveiling Linear Mode Connectivity of Re-basin from Neuron Distribution Perspective"
_ICLR.cc/2024/Conference — Submitted to ICLR 2024_

### Official Review · Reviewer_ehnr · 2023-10-26

**Soundness:** 2 fair
**Presentation:** 2 fair
**Contribution:** 2 fair
**Rating:** 3
**Confidence:** 5

**Summary:**

This paper connects the theory of random matching problem and the weight matching method of Git Re-basin (Ainsworth et al). They give an upper bound over the loss barrier between two models and the bound is related to the distribution of model parameters (which could be formulated as discrete entropy). Also, they find pruning could potentially decrease the loss barrier between two models after re-basining.

**Strengths:**

- First of all, the authors identified an important problem that is why the permutation methods (including weight matching proposed in Ainsworth et al., 2022 and Entezari et al., 2021) could achieve LMC between two independently trained models and why they fail in some cases (e.g. early training).
- A good point of this paper is to build the connection between random matching problem and loss barrier between two models.
- Also, the phenomenon that pruning could potentially decrease the loss barrier after re-basining is interesting.

**Weaknesses:**

Theoretical side:
The theorem 3.1 tells a simple conclusion that is the loss barrier between models are bounded by the distance between their model parameters, which is intuitive and aligns with most literatures (I am not questioning on theorem 3.1). The theorem 3.2 gives an upper bound over the distance between two random model parameters. After that, theorem 3.3 just combines above two theorems. Therefore, I thought the core part of this paper is simply the theorem 3.2 (sometimes the author refers it as Lemma 3.2, which might be a typo...).
- However, in a real case, the model parameters might not be "random" and the bound given by theorem 3.2 might be useless in practice. In that case, the relation between the loss barrier and the entropy could be overestimated.
- Also, the theorem 3.2 actually directly comes from the random Euclidean matching problems (as mentioned by the authors in Sec 3.)
Above all, the theoretical contribution of this paper is marginal.

Experimental side:
1. Sec 4.2, the experiments are quite "toy" (Polynomial Task, single output MLP and only first layer are tested). For both Sec 4.1 & 4.3, the experiments are conducted over standard image classification task and models, however, for Sec 4.2, the experiments are quite trivial. Harder datasets and models are needed for Sec 4.2.
2. Still 4.2, only the change of distribution of model parameters before training and after training cannot show a strong correlation between the loss barrier and entropy. More carefully designed experiments are needed.
3. Pruning experiments are interesting but the explanation of why Only Pruning and Lottery Ticket Hypothesis fail is not that clear. Actually, the failure cannot be predicted by their theoretical analysis.
4. From Figure 5, the loss does not always first decrease and then increase. For MLP and CIFAR-10, the loss first increases with pruning ratio actually. The phenomenon contradicts with the results of other dataset and models. Also, the loss curves of train dataset and test set are not always consistent.
5. The entropy before and after pruning should be presented for comparison.

Overall, the experimental contribution is not sound to me.

**Questions:**

1. For Sec 4.1, I wonder if all the models are all randomly initialized, how could loss barrier exist? Because in my mind, one randomly initialized model could be a "random guess" classifier, and therefore, if two "random guess" classifier are interpolated, the interpolated model should still random guess, then there couldn't exist any loss barrier.

---

### Official Review · Reviewer_Lp2L · 2023-10-28

**Soundness:** 2 fair
**Presentation:** 2 fair
**Contribution:** 2 fair
**Rating:** 5
**Confidence:** 3

**Summary:**

This paper investigates the theory of re-basins, explores when and why re-basins improve linear mode connectivity, and examines the problem from a neuron distribution perspective. The authors conducted analytical experiments on neuron distributions with different initializations, comparisons before and after fine-tuning, pruning, and more. Notably, the pruning-then-fine-tuning experiments yield interesting findings. Finally, the authors demonstrate how to apply their theory to other methods, such as OTFusion and FedMA.

**Strengths:**

- This paper examines linear mode connectivity (LMC) after re-basin through changes in neuron distribution.
- The authors provide both theoretical analysis as well as practical experiments.
- The finding that pruning and then fine-tuning at a higher rate improves re-basin is an interesting discovery.
- The writing is clear.

**Weaknesses:**

The analysis using entropy is rather trivial in hindsight; for instance, different initializations result in a higher entropy of the neuron distribution (Fig. 2), and training changes the neuron distribution from uniform to bi-modal (Fig. 3). Overall, the theoretical/analytical contribution may be somewhat thin.

**Questions:**

N/A

---

### Official Review · Reviewer_3GTP · 2023-10-31

**Soundness:** 2 fair
**Presentation:** 1 poor
**Contribution:** 2 fair
**Rating:** 3
**Confidence:** 4

**Summary:**

The authors aim at their research to explain the linear mode connectivity, i.e., the ability of neural networks to be connected by a low loss line, through the properties of the weights values distribution. In particular they are using discrete Shannon entropy of the weights distribution as a characteristic that should be low in order to allow for low barrier between models. They provide a theoretical result for the upper bound on the difference of network outputs that depends on the entropy of the weights distribution. Further they evaluate empirically the values of the barrier and entropy for the newly initialized models, after training and after pruning. It is proposed to use pruning as a method for enhancement of the linear connectivity via empirical results with some state-of-the-art vision models and fusion methods.

**Strengths:**

Paper aims at having a theoretically justified result about effect of entropy on the barrier between two models. Additionally, this result is used to propose an applicational enhancement for permutation based matching between networks in order to fuse them (as a fusion ensemble or federated learning).

**Weaknesses:**

Paper is hard to follow and there are many questions that arise upon reading it.

The first question that is aimed to answer is about hardness of LMC at initialization as in Ainsworth et al. The answer that paper gives is that it depends on the entropy of the initialization, but there is no correspondence to the initial question in a sense of considering same experiments as in Ainsworth et al. and changing initialization to less entropy one. Moreover, already at Ainsworth et al. it is mentioned that in a concurrent work it was shown that mapping found after training improves LMC at initialization. How can entropy explain this? In Entezari et al. it was proven that for wide networks LMC is possible at initialization. How does this result connects to the entropy?

The authors use definition of the barrier introduced in Entezari et al. Nevertheless, it is not quite clear to me how \alpha is selected there (and later in the paper Entezari et al. just use 1/2, thus returning to the classical definition of Frankle et al.). I wonder how the authors use this definition. Moreover, in the theorem 3.1 instead of bounding barrier, some other value is bounded - supremum over difference between network output for interpolated weights and interpolation of network outputs. This already does not correspond to any of the definitions. I can assume that they still can be linked, but this linking is not given in the paper.

Proof of theorem 3.1 does not require permutations per se - it is introduced in the very end to shrink the barrier. So overall, it should mean that entropy of neuron distribution can bound the difference between output of interpolated model and interpolation of the models? Moreover, result in lemma 3.2 requires 0 mean of the distribution of the neuron weights - I think it does not have to be the case in a trained network. Finally, the proof of theorem 3.3 is absent. While it is connecting theorem 3.1 and lemma 3.2 it requires some polynomial properties of the network, which are not explained.

The experiment in section 4.2 uses a network with sigmoid activation. Such activation is known to bin into two corner cases with training (see discussion about experiments in the work on information bottleneck of N.Tishby). Therefore the demonstrated peaks in the values of the weights does not necessarily mean the desired result.

The argument for why pruning by itself does not result in decrease of the barrier value is not convincing for me. The value of the barrier is normalized by the losses of the two initial models, therefore it should not mean that the barrier will be high if models do not perform well. And fine-tuning after matching is known to always improve LMC, so once again this experiment does not show the desired result.

Minor:

- currently, pruning does not smoothly integrate in the paper: in the introduction it is formulated as "we observe that pruning can improve LMC, but what is the mechanism", while in the paper itself pruning is directly proposed as a method to decrease entropy.

- the Figure 1 is very sloppy - why the shadow areas are exactly where solutions would be? How is it justified?

- the formulation "making neuron distribution gather around minima" is unclear to me

- in the introduction it is controversially claimed that the proposed method makes re-basin easier, while it is not the case

- neuron entropy and neural entropy are used in parallel in the paper, while neuron entropy is the term introduced

- lemma 3.2 is called theorem 3.2 in the text

- using assumptions from another paper (Mei et al.) without explaining them (at least in appendix) makes the paper not self-contained

**Questions:**

Please see questions in the section "Weaknesses".

---

> ### Comment · Reviewer_3GTP · 2023-11-23
>
> Unfortunately I do not find answers to my questions in the general responses, so I stay with my score.

---

### Official Review · Reviewer_y1Dw · 2023-11-01

**Soundness:** 1 poor
**Presentation:** 2 fair
**Contribution:** 1 poor
**Rating:** 3
**Confidence:** 3

**Summary:**

The paper proposes a conjecture that behavior of loss barrier in Linear Mode Connectivity (LMC) of Deep Neural Networks can be explained through entropy (non-uniformity) of distribution imposed on the neurons of the network. The authors show that for multi layered perceptron the loss barrier between two networks are upper bounded by O(polynomial(number of neurons in hidden layers) x exponential(entropy of neuron distributions)). The paper goes on the explain that pruning results in reduction in entropy and hence loss barrier of LMC for pruned networks are better, resulting in higher accuracy of fused model. The experiments in the paper are two fold (a) to show empirically that entropy and LMC are connected (b) apply pruning to various existing methods for better fusion of deep networks.

**Strengths:**

The ideas presented in the paper are interesting and could have potential applications in understanding of LMC, as well as in improving accuracy of model fusion methods.

a) The connection between loss barrier of LMC and entropy is novel and should be explored further.

b) The application of pruning in model fusion seems to be a positive direction in improving the accuracy of fused model.

**Weaknesses:**

In the current state the results in the paper does not seem to back the contributions mentioned in the introduction and abstract. There are various weak points:
1) The paper proposes a conjecture on how entropy could explain some of the behavior in loss barrier and LMC. However various parts of abstract and introduction tries to hint that the proposal contains a "theory to demystify the mechanism behind LMC of re-basin". The contributions needs to be reworded.
2) The paper deals only with MLP with bias=0. The presence of bias term and cases like Deep networks with transformer units seem to be non-trivial to extend to as opposed to as claimed in the paper.
3) The theorem provides an upper bound on loss barrier and entropy of imposed distributions on neurons. However two things seem to be missing. The first is usefulness of the bound. The polynomial function in bound is over width of NNs across all the depth. This number could be potentially very large rendering the bound trivial. It has to be shown that the entropy term is significant over the other terms in the bound to be able to attribute the behavior of LMC to entropy. Secondly, there is hardly any section devoted to the nature of such distributions that would be present in deep networks after they are trained. Both of these seem to be crucial to the claims made.
4) There seems to be a confusion about if the results in paper are about better LMC (higher accuracy along linear path) vs smaller loss barrier in the presented material. I think the theorems need to be re-interpreted for converged NNs.
5) The experiment results do not contain any standard deviations for most parts. This makes it hard to understand if the benefits from proposed pruning are significant in nature. The results in table 2 and those of FedMA are very close that it can be considered statistically significant.
6) Results on FedMA are not included in the main paper. Please replace some section of Table 2 to accommodate the same.
7) Based on the main theorem proposed, section 4.2 argues that pruning results in lower entropy and hence better loss barrier. However the later part of the section claims that "LMC first increases and then drops Figure 5". This seems to be not explained by the theory proposed and is a major point of confusion in presented material. What is meaning of LMC increasing or decreasing? Is this referring to loss barrier or accuracy along linear mode?

**Questions:**

Q1) Could you provide the comparison between the relative order of magnitudes of various constants in the presented Theorem 3.1? It seems crucial to understand the same to understand the impact of entropy in that equation.

Q2) Consider two untrained VGG network on MNIST dataset vs two trained VGG network. As seen in Figure 4, the trained networks have non-trivial loss barrier. But the non-trained networks would have closer to zero loss barrier because they already are at a high point in the loss landscape. There can be different ways in which this can be made to happen. However the entropy of trained networks are lower, but it does not explain the different in loss barrier? This seems contradictory to the result in the paper.

Q3) I don't seem to get grasp on how bias could be added to weight matrix through small adjustment as mentioned just before section 2.2. Please elaborate the same in the light of if equation 3 still holds.

Q4) In experiment 4.1, why are the curves later fit to the observation vs plotting the entropy exp(.) function.

Q5) Are bias terms = 0 in all the networks considered in experiments? If not how is OTFusion extended for the same.

Q6) Could you please why LTH does not lead to good fused networks? Different matching algorithms should be able to combine two models generated by LTH.

Q7) What does '/' mean in Table 2? Please also include base model accuracy for fusion. Why are models fine tuned till epoch 30?

Typos:

a) Please fix VGG11 vs VGG16 in Table 2 and section 5. They seem to be interchangeably used.

---

### Author Response · Authors · 2023-11-23
**General Response_6 (Response to the most misunderstanding parts)**

> 5. Doubts about our theoretical novelty (for example, statements like "Theorem 3.1 presents a straightforward conclusion" or "Theorem 3.2 appears to directly stem from random Euclidean matching problems").
>

Our theoretical contributions exhibit several novelties:

(1) We propose the first theorem(Theorem 3.1) providing a theoretical analysis of LMC in multi-layer MLPs.
(2) We propose the first theorem(Theorem 3.2) considering the impact of neuron parameter distribution on LMC, beyond the influence of network architecture on LMC in previous work.
(3) We present the first theoretical analysis of LMC in trained neural networks, taking advantage of  the perspective of neuron parameter distribution.
(4) We analyze how pruning can enhance LMC, based on the relationship between LMC and the entropy of neuron parameter distribution and our Lemma 4.1, which shows pruning can reduce the entropy of neuron parameter distribution.

Additionally, the complete proofs of our theorems provided in the appendix reveal that the process of proof is intricate and the content of the theorems themselves is not simple. While some aspects of our theorems might align with intuition, it doesn't imply that our theoretical results lead to a "simple conclusion."



---

Thanks again for the valuable comments. We will make revisions to our paper based on your suggestions.

---
[1] Entezari R, Sedghi H, Saukh O, et al. The Role of Permutation Invariance in Linear Mode Connectivity of Neural Networks[C]//International Conference on Learning Representations. 2022.

[2] Ainsworth S, Hayase J, Srinivasa S. Git Re-Basin: Merging Models modulo Permutation Symmetries[C]//The Eleventh International Conference on Learning Representations. 2023.

[3] Song Mei, Andrea Montanari, and Phan-Minh Nguyen. A mean field view of the landscape of
two-layer neural networks. Proceedings of the National Academy of Sciences, 115(33):E7665–
E7671, 2018. doi: 10.1073/pnas.1806579115. URL https://www.pnas.org/doi/abs/
10.1073/pnas.1806579115.

[4] Song Mei, Theodor Misiakiewicz, and Andrea Montanari. Mean-field theory of two-layers neural
networks: dimension-free bounds and kernel limit. In Conference on Learning Theory, pp.
2388–2464. PMLR, 2019.

---

### Author Response · Authors · 2023-11-23
**General Response_5 (Response to the most misunderstanding parts)**

>
4. The experimental design for validating the relationship between the enhancement of LMC and entropy decrease after training in Section 4.2 is too simple and lacks validation using the experimental settings for LMC enhancement after training in [2].
>

That's because to address this issue, we would need to statistically compute the entropy of the neuron distribution of layers in models described by Ainsworth et al. This is a rather challenging task. If, for a one-dimensional random variable, we need $n$ observations to fairly accurately estimate its entropy, then for a $k$-dimensional random vector, we might require $n^k$ observations for a reasonable estimation of its entropy. In [2], the hidden layer neurons are mostly 32 to 64-dimensional random vectors. Estimating the entropy of such random vectors poses a significant challenge (needing $n^{32}$ observations, yet training a single hidden layer provides only 32 neurons, perhaps offering just 32 observations, hence requiring $\frac{n^{32}}{32}$ trainings) to gather sufficient observation information for an entropy estimation. It's for this reason that in the paper, we opted for a simpler task similar to that used in [3]. In this task, because the neurons in the first hidden layer are one-dimensional, it's easier to straightforwardly estimate their distribution entropy.

In subsequent revisions, we will work on addressing the issue of statistical estimation of entropy for high-dimensional random variables. In Section 4.2, we will design more complex experiments and validate them using experimental settings demonstrating the enhanced LMC after training as outlined in [2].

---
[1] Entezari R, Sedghi H, Saukh O, et al. The Role of Permutation Invariance in Linear Mode Connectivity of Neural Networks[C]//International Conference on Learning Representations. 2022.

[2] Ainsworth S, Hayase J, Srinivasa S. Git Re-Basin: Merging Models modulo Permutation Symmetries[C]//The Eleventh International Conference on Learning Representations. 2023.

[3] Song Mei, Andrea Montanari, and Phan-Minh Nguyen. A mean field view of the landscape of
two-layer neural networks. Proceedings of the National Academy of Sciences, 115(33):E7665–
E7671, 2018. doi: 10.1073/pnas.1806579115. URL https://www.pnas.org/doi/abs/
10.1073/pnas.1806579115.

[4] Song Mei, Theodor Misiakiewicz, and Andrea Montanari. Mean-field theory of two-layers neural
networks: dimension-free bounds and kernel limit. In Conference on Learning Theory, pp.
2388–2464. PMLR, 2019.

---

### Author Response · Authors · 2023-11-23
**General Response_4 (Response to the most misunderstanding parts)**

>    3. When initialization, if all the models are randomly initialized, how could a loss barrier exist? Because the non-trained networks tend to have closer to zero loss barriers because they are already at a high point in the loss landscape.
>

Previous work has demonstrated the existence of a loss barrier during random initialization：Firstly, the experimental evidence can be found in Figure 3 of [2], which demonstrates the existence of the loss barrier before training and it is challenging to reduce using the re-basin method. Figure 3, from an experimental perspective, illustrates that LMC enhances after training (indicating relatively large loss barriers when initialization). Secondly, [1]'s Theorem 3.1 provides theoretical insights into the magnitude of loss barriers during initialization.

When randomly initializing, not all initialized parameter points lie at a relative high point in the loss landscape. While any two untrained points selected on the loss landscape might be at a relative high point, there is still a possibility of higher points along their line connection. Since these two points are chosen arbitrarily, there's a considerable chance they might not reside at a high point on the loss landscape.
 While picking points on the loss landscape, although it's possible to select high points, there's still a substantial probability of choosing non-high points. Simultaneously, there's a considerable probability of higher points existing along their connecting line. Hence, it's not accurate to assume that parameter points at initialization have a high loss and then there are no higher points along the line connecting them causing a large loss barrier.

Additionally, we cannot simply assume that "one randomly initialized model could be a 'random guess' classifier, and therefore, if two 'random guess' classifiers are interpolated, the interpolated model should still be a random guess, and thus, there couldn't exist any loss barrier." This is because, based on our definition of the Loss Barrier, it represents the maximum difference along a path, and the value of the interpolated model corresponding to this maximum difference is not entirely random. There's a computation involved in determining this maximum value.

---
[1] Entezari R, Sedghi H, Saukh O, et al. The Role of Permutation Invariance in Linear Mode Connectivity of Neural Networks[C]//International Conference on Learning Representations. 2022.

[2] Ainsworth S, Hayase J, Srinivasa S. Git Re-Basin: Merging Models modulo Permutation Symmetries[C]//The Eleventh International Conference on Learning Representations. 2023.

[3] Song Mei, Andrea Montanari, and Phan-Minh Nguyen. A mean field view of the landscape of
two-layer neural networks. Proceedings of the National Academy of Sciences, 115(33):E7665–
E7671, 2018. doi: 10.1073/pnas.1806579115. URL https://www.pnas.org/doi/abs/
10.1073/pnas.1806579115.

[4] Song Mei, Theodor Misiakiewicz, and Andrea Montanari. Mean-field theory of two-layers neural
networks: dimension-free bounds and kernel limit. In Conference on Learning Theory, pp.
2388–2464. PMLR, 2019.

---

### Author Response · Authors · 2023-11-23
**General Response_3 (Response to the most misunderstanding parts)**

>    2. Theoretical proofs are suitable for explaining LMC at initialization but not suitable for explaining LMC after training (e.g., "I think the theorems need to be re-interpreted for converged NNs.")
>

In the context of the LMC topic, this is a frequently asked question. Currently, due to the complexity of theory, theoretical analyses of LMC are primarily confined to analyzing networks at their initialization [1]. So far, there have been no theoretical analyses of LMC after training (or converged NNs). The approach taken by researchers is to provide theoretical proofs for initialization while relying on experiments to demonstrate results for converged NNs.

**And this paper actually marks the first breakthrough on this matter, providing the initial theoretical analysis of LMC after training (or converged NNs).** The discussion can be found in section 4.2. The approach adopted in this paper for analyzing LMC after training is as follows:

1. Theorems 3.1 and 3.2 hold true for networks where parameters of neurons within the same layer follow an arbitrary distribution i.i.d. as random vectors.
2. In Mean Field Theory, for neural networks that satisfy conditions A1 to A4, the parameters of each neuron within the same layer are considered as random vectors that follow an independent and identically distribution during the training process [3,4].
3. Based on 1 and 2, for neural networks that fulfill conditions A1 to A4 in [3,4] during training, the analysis of their LMC can also be described using Theorems 3.1 and 3.2.


Building upon this line of thinking, this paper explains why LMC strengthens after training. This occurs because both LMC before training and after training of neural networks can be described using Theorem 3.2. Theorem 3.2 establishes that neural networks with lower distribution entropy of neuron parameters exhibit stronger LMC. However, as the network becomes increasingly deterministic after training, the entropy of distribution of neuron parameters decreases after training. Consequently, the LMC of the network strengthens after training.

---
[1] Entezari R, Sedghi H, Saukh O, et al. The Role of Permutation Invariance in Linear Mode Connectivity of Neural Networks[C]//International Conference on Learning Representations. 2022.

[2] Ainsworth S, Hayase J, Srinivasa S. Git Re-Basin: Merging Models modulo Permutation Symmetries[C]//The Eleventh International Conference on Learning Representations. 2023.

[3] Song Mei, Andrea Montanari, and Phan-Minh Nguyen. A mean field view of the landscape of
two-layer neural networks. Proceedings of the National Academy of Sciences, 115(33):E7665–
E7671, 2018. doi: 10.1073/pnas.1806579115. URL https://www.pnas.org/doi/abs/
10.1073/pnas.1806579115.

[4] Song Mei, Theodor Misiakiewicz, and Andrea Montanari. Mean-field theory of two-layers neural
networks: dimension-free bounds and kernel limit. In Conference on Learning Theory, pp.
2388–2464. PMLR, 2019.

---

### Author Response · Authors · 2023-11-23
**General Response_2 (Response to the most misunderstanding parts)**

>1. There is no detailed discussion of network architecture parameters (including network width) on the relationship of LMC, for example, (1) the impact analysis of width on LMC has not been conducted, (2) without considering the magnitude of the impact of network width on LMC, it is not possible to determine whether the distribution of neurons affects LMC.
>
We have the following explanation:
(1) Understanding the theorem(Theorem 3.1 and Theorem 3.2) in this article first requires an understanding of objective of this article, which is to "measure the impact of different neuron parameter distributions on LMC after re-basing." In this process, the network structure is fixed, meaning that parameters such as network width remain constant and do not affect the discussion in this article. Our aim is to discuss how altering neuron distribution affects LMC when the network structure is fixed.

(2) Secondly, regarding the doubt about the comparison between magnitudes of network width and entropy of neuron parameter distribution, in Theorems 3.1 and 3.2, it can be observed that the relationship between network structure-related quantities and distribution-related quantities (i.e., entropy) is multiplicative. In other words, in these theorems, the bound of LMC is actually expressed in the following form: $A_{arch}B_{distribution}$ (where $A_{arch}$ represents network structure-related quantity, and $B_{distribution}$ represents distribution-related quantity). Regardless of the magnitude of $A_{arch}$, given its fixed value, the relative impact of $B_{distribution}$ on LMC remains unaffected.

Based on (1) and (2), it is no need for us to conduct an analysis of the magnitude of network structure-related quantities in the paper because their specific magnitudes do not impact our discussion.

---
[1] Entezari R, Sedghi H, Saukh O, et al. The Role of Permutation Invariance in Linear Mode Connectivity of Neural Networks[C]//International Conference on Learning Representations. 2022.

[2] Ainsworth S, Hayase J, Srinivasa S. Git Re-Basin: Merging Models modulo Permutation Symmetries[C]//The Eleventh International Conference on Learning Representations. 2023.

[3] Song Mei, Andrea Montanari, and Phan-Minh Nguyen. A mean field view of the landscape of
two-layer neural networks. Proceedings of the National Academy of Sciences, 115(33):E7665–
E7671, 2018. doi: 10.1073/pnas.1806579115. URL https://www.pnas.org/doi/abs/
10.1073/pnas.1806579115.

[4] Song Mei, Theodor Misiakiewicz, and Andrea Montanari. Mean-field theory of two-layers neural
networks: dimension-free bounds and kernel limit. In Conference on Learning Theory, pp.
2388–2464. PMLR, 2019.

---

### Author Response · Authors · 2023-11-23
**General Response_1**

We thank the reviewers for their valuable comments and precious time. We find the reviewers' comments highly helpful for improving our paper, and we will incorporated them into our revised paper.

Additionally, we've found that due to the rushed writing of our paper, reviewers have had certain misunderstandings about the perspectives and content of our paper. Therefore,  we will first clarify our contributions and then address the areas where reviewers have had the most misconceptions regarding our paper.

---
### Clarifying the contributions

We would like to kindly clarify our contributions as follows.  **1)** Previous research on LMC primarily focuses on the impact of neural network structure (for example, the width and depth of neural networks) on LMC [1][2].  We establish a theoretical connection between neural network LMC after re-basin and the random Euclidean matching problems, presenting theoretical results in Theorems 3.1 and 3.2. **These theorems demonstrate the joint impact of neural network structure and neuron distribution on LMC, marking the first theoretical consideration of neuron distribution's influence on LMC**. Additionally, Theorem 3.1 is the **first analysis of LMC after re-basin in multi-layer MLPs** and the previous theorem in [1] only analyzes two-layer MLP.  **2)** Using the relationship between neuron distribution's entropy and LMC described in Theorem 3.2, we utilized mean field theory [3][4] to **analyze LMC after re-basin for neural network after training for the first time**. This analysis provide the first explanation that training enhances neural network LMC for training processes satisfying the A1~A4 assumptions of mean field theory. **3)** We demonstrated that pruning reduces the entropy of neuron distribution(through Lemma 4.1) and, leveraging the relationship between neuron distribution's entropy from Theorem 3.2 and LMC after rebasing, **we explain how pruning can enhance LMC after rebasing in neural networks. Through experiments, we validated the effectiveness of pruning in improving LMC after rebasing.** **4)** We applied the method of enhancing LMC post-rebasing through pruning to neuron-matching-based ensemble methods such as OT-fusion and federated learning's FedMA method. **Our findings reveal that our pruning method can improve the result of model fusion in OT-fusion and FedMA**

---
[1] Entezari R, Sedghi H, Saukh O, et al. The Role of Permutation Invariance in Linear Mode Connectivity of Neural Networks[C]//International Conference on Learning Representations. 2022.

[2] Ainsworth S, Hayase J, Srinivasa S. Git Re-Basin: Merging Models modulo Permutation Symmetries[C]//The Eleventh International Conference on Learning Representations. 2023.

[3] Song Mei, Andrea Montanari, and Phan-Minh Nguyen. A mean field view of the landscape of
two-layer neural networks. Proceedings of the National Academy of Sciences, 115(33):E7665–
E7671, 2018. doi: 10.1073/pnas.1806579115. URL https://www.pnas.org/doi/abs/
10.1073/pnas.1806579115.

[4] Song Mei, Theodor Misiakiewicz, and Andrea Montanari. Mean-field theory of two-layers neural
networks: dimension-free bounds and kernel limit. In Conference on Learning Theory, pp.
2388–2464. PMLR, 2019.

---

### Meta-Review · Area_Chair_AUnK · 2023-12-05

**Metareview:**

The paper considers the linear mode connectivity with re-basin methods from the perspective of neuron distribution. In particular, the authors connect the LMC to the entropy of the neuron distribution and provide theoretical justification for why LMC of re-basin changes during training, how it depends on initialization, and how it interacts with pruning.

## Strengths

- Better understanding LMC and re-basin methods is both very interesting scientifically and could be very practically impactful
- Multiple reviewers pointed out that the observations about the interaction of re-basin LMC and pruning are novel and interesting
- The idea of considering neuron entropy and random matching problems is interesting, as was pointed out by several reviewers

## Weaknesses

- Generally, the reviewers were not convinced by the evidence presented in the paper, and were not convinced that the proposed theory explains the empirical observations

**Justification For Why Not Higher Score:**

The reviewers vote unanimously in favor of rejecting the paper, and remain unconvinced by the rebuttal. While the paper presents interesting new ideas or studying LMC with re-basin, the results were not convincing and the authors did not provide sufficient empirical support to show that the theory provides an explanation for the practical observations.

**Justification For Why Not Lower Score:**

N/A

---

### Decision · Program_Chairs · 2024-01-16

Reject